# Nacα protects the larval fat body from cell death by maintaining cellular proteostasis in *Drosophila*

Takayuki Yamada[1], Yuto Yoshinari[2], Masayuki Tobo[2], Okiko Habara[1] & Takashi Nishimura [1,2] ✉

Protein homeostasis (proteostasis) is crucial for the maintenance of cellular homeostasis. Impairment of proteostasis activates proteotoxic and unfolded protein response pathways to resolve cellular stress or induce apoptosis in damaged cells. However, the responses of individual tissues to proteotoxic stress and evoking cell death program have not been extensively explored in vivo. Here, we show that a reduction in Nascent polypeptide-associated complex protein alpha subunit (Nacα) specifically and progressively induces cell death in *Drosophila* fat body cells. *Nacα* mutants disrupt both ER integrity and the proteasomal degradation system, resulting in caspase activation through JNK and p53. Although forced activation of the JNK and p53 pathways was insufficient to induce cell death in the fat body, the reduction of *Nacα* sensitized fat body cells to intrinsic and environmental stresses. Reducing overall protein synthesis by mTor inhibition or *Minute* mutants alleviated the cell death phenotype in *Nacα* mutant fat body cells. Our work revealed that Nacα is crucial for protecting the fat body from cell death by maintaining cellular proteostasis, thus demonstrating the coexistence of a unique vulnerability and cell death resistance in the fat body.

The correct localization and function of cellular proteins are vital for maintaining cellular homeostasis. Protein homeostasis (proteostasis) is primarily affected by a series of biological pathways, including protein biosynthesis, folding, trafficking, and degradation[1,2]. Furthermore, living cells are subjected to various intrinsic and environmental challenges, such as reactive oxygen species (ROS) and ultraviolet (UV) irradiation, which cause various types of protein damage. Thus, the careful orchestration of stress sensors and adaptive signaling pathways is crucial for maintaining proteostasis in each subcellular compartment. In agreement with this, the functional decline in proteostasis is causally related to human diseases, some of which are present at birth but mostly upon aging[3–5].

The heat shock, or proteotoxic stress, response (HSR/PSR) regulates cytosolic proteostasis by inducing the expression of cytosolic molecular chaperones, such as heat shock proteins (Hsps)[6,7]. Furthermore, two mechanisms, namely the proteasome and autophagy, ensure cytosolic proteostasis by regulating proteolysis. The ubiquitin-proteasome system (UPS) is involved in the proteolysis of short-lived functional proteins and abnormal proteins with aberrant structures. Autophagy also functions in proteolysis through the selective or non-selective engulfment of cytoplasmic constituents[8,9]. As excess autophagy induces cell death under certain conditions[10], the accumulation of autophagosomes often coincides with cell death[11]. Similarly, ectopic proteasome impairment or reduction in the UPS facilitates cell death[12].

Most secretory and membrane proteins are folded in the endoplasmic reticulum (ER) by ER-localized chaperones and protein disulfide isomerases. ER stress occurs under various physiological and pathological conditions when protein-folding capacity is

[1]Laboratory for Growth Control Signaling, RIKEN Center for Biosystems Dynamics Research (BDR), Kobe, Hyogo 650-0047, Japan. [2]Laboratory of Metabolic Regulation and Genetics, Institute for Molecular and Cellular Regulation, Gunma University, Maebashi, Gunma 371-8512, Japan. ✉e-mail: t-nishimura@gunma-u.ac.jp

overwhelmed. ER stress triggers the ER to nucleus signaling pathway called the unfolded protein response (UPR), which induces the synthesis of ER chaperones and reduces global protein synthesis[13,14]. An evolutionarily conserved UPR pathway is mediated by the ER-resident RNase IRE1 and its target Xbp1 transcription factor. However, if ER stress is extensive or ER function cannot be restored, ER stress triggers caspase activation, resulting in cell death. In the fruit fly *Drosophila*, tissues with high protein secretory load are associated with inherent ER stress, as revealed by IRE1 activity and basal Xbp1 expression[15,16], suggesting that adaptive stress responses occur during development in a tissue- and stage-specific manner. Notably, ER stress is closely linked to cytosolic proteostasis, as misfolded proteins in the ER are retro-translocated to the cytoplasm for ubiquitin-mediated degradation, which is referred to as the ER-associated degradation (ERAD) pathway[17,18].

The nascent polypeptide-associated complex (NAC), composed of an alpha subunit (Nacα) and a beta subunit (Nacβ), is an evolutionarily conserved and ubiquitously expressed protein essential for organismal viability. NAC binds to ribosome-associated nascent polypeptides and competes with the signal recognition particle (SRP) to prevent mistargeting of cytosolic and mitochondrial proteins to the ER, thereby protecting against ER stress[19,20]. In several species and cell lines, the loss of NAC activates the ER stress response, which eventually leads to cell death through JNK and caspase activation[21]. Thus, NAC plays a crucial role in protein quality control and the maintenance of cellular homeostasis. The molecular mechanisms underlying stress-induced signaling pathways and the crosstalk between cytosolic and ER proteostasis have been well-documented in several organisms. However, how each tissue responds to proteotoxic stress and evokes a transition from adaptive to cell death program has not been thoroughly explored in vivo.

Interestingly, cell ploidy may influence cell death and survival. Most somatic cells in eukaryotes are diploid and contain two sets of chromosomes, whereas some cells are polyploid due to endoreplication without mitosis, which leads to an elevated number of chromosomes[22,23]. Endoreplicated cells are relatively restricted in mammals but are more common in invertebrates[24]. In *Drosophila*, most larval-specific tissues, such as the salivary glands and fat bodies, are polyploid[25]. As cell ploidy generally correlates with cell size and protein production capacity, endoreplication is assumed to be functionally important for secretion, nourishment, and structural protection. In both mammals and *Drosophila*, some polyploid cells are resistant to cell death due to genotoxic stress, which induces apoptosis in mitotic cells[26–28]. However, the response of polyploid cells to other cellular stresses, including a decline in the proteostasis network, has not been systematically analyzed.

In this study, we showed that the *Drosophila* larval fat body is resistant to cell death not only by DNA stress but also by several cellular stresses, including ER stress. In contrast, we found that a hypomorph mutant of *Nacα* induces sporadic cell death in the larval fat body. Remarkably, the cell death phenotype is specific to the fat body and progressive during development. Our study revealed that the larval fat body is a vulnerable organ that induces a cell death program through proteostasis impairment.

## Results

### The larval fat body is resistant to cell death by several cellular stresses

The fat body of insects is equivalent to that of mammalian liver and adipocytes[29,30]. The larval fat body is resistant to genotoxic stress due to reduced protein levels of the tumor suppressor p53 and chromatin silencing of pro-apoptotic genes[31]. However, paradoxically, the overexpression of pro-apoptotic genes fails to induce cell death in the fat body shortly after transient induction[26,32]. Therefore, we aimed to re-evaluate the effects of pro-apoptotic genes in the larval fat body

(Fig. 1a). To avoid possible effects on fat body development, we transiently induced target genes in the fat body at the mid-third instar using a *Cg-Gal4* driver combined with a thermosensitive Gal4 repressor *Gal80ts*. We found that pro-apoptotic genes (*rpr*, *hid*, and *grim*) increased cleaved Death-caspase-1 (cDcp1) staining 24 h after the temperature shift (Fig. 1b, c). Consistently, transient knockdown of the anti-apoptotic protein *Diap1*, which inhibits caspase activity, increased cDcp1 signals (Supplementary Fig. 1a). cDcp1-positive cells had condensed nuclei and reduced cortical F-actin levels, suggesting that these cells undergo caspase-dependent cell death. Furthermore, the induction of pro-apoptotic genes and knockdown of *Diap1* showed propidium iodide (PI) staining, indicative of late cell death (Supplementary Fig. 1b). Interestingly, cell death occurred in a mosaic manner, although *Cg-Gal4* transcriptional activity and *Diap1* expression were almost equally detectable in fat body cells, as revealed by the dual-color fluorescent reporter TransTimer[33] and a *Diap1-lacZ* reporter, respectively (Fig. 1d and Supplementary Fig. 1c). Given that these pro-apoptotic genes induce cell death in imaginal discs within hours after induction[26,31,34], cell death processes in the fat body may take longer than that in the imaginal discs.

Besides DNA stress, it is well known that oncogenic and ER stresses readily induce apoptosis in imaginal discs[34–36]. However, it remains unclear whether these cellular stresses induce cell death in the fat body. To address this issue, we compared the cell death phenotypes of fat bodies and imaginal discs. Gene induction in the imaginal discs was performed using a *nub-Gal4* driver expressed in the developing wing pouch. We found that transient induction of a constitutively active form of *Src42A* oncogene (*Src42A-CA*) or JNK-kinase *hemipterous* (*hep-CA*) failed to induce fat body cell death, although these constructs strongly induced cell death in wing discs (Fig. 1e, f). *Src42A-CA* notably increased cortical F-actin staining in the fat body, presumably by regulating the cortical actin network[37]. Similarly, the induction of ER stress by either a dominant-negative form of the ER chaperone *Hsc70-3* (*Grp78/Bip* in *Drosophila*) or a long wild-type isoform of *Presenilin* (*Psn + 14*) had no effect on the fat body 24 h after induction (Fig. 1e). These constructs induced apoptosis in wing discs (Fig. 1f), as reported previously[34,38]. Disruption of cytosolic proteostasis through a temperature-sensitive, dominant-negative form of proteasome subunits *Prosβ2* and *Prosβ6* (*Prosβ-DN*) had no effect on the fat body but induced cell death in wing discs, although the phenotype was relatively mild. Consistent with previous observations[26], *p53* overexpression strongly induced cell death in wing discs, but not in fat bodies. Collectively, these results suggest that the larval fat body is resistant to cell death by DNA stress as well as several cellular stresses, including oncogenic and ER stresses.

### Dysfunction of *Nacα* causes sporadic fat body cell death in *Drosophila* larvae

During the experiments, we serendipitously found that the deletion mutant allele of the sugar-responsive transcription factor *sugarbabe* (*sug*), named *sug174*[39], resulted in fat body cells with small nuclei in the third instar larvae (Fig. 2a). These sporadic small cells were positive for cleaved active caspase (Cas3) and were considered caspase-associated cell death (Fig. 2b). However, transheterozygotes of *sug174* mutants over a deficiency allele (*Df(2R)Exel7123*, hereafter referred to as *sug-Df*) had no phenotype, indicating that the cell death phenotype is independent of *sug* (Fig. 2c).

The cell death phenotype of the homozygous mutant fat body prompted us to identify the causative gene. Genetic mapping of the mutation was performed using available deficiency kits. We identified a small region near the *sug* locus that contained eight genes and a long non-coding RNA (Fig. 2c, d and Supplementary Fig. 1d). We further screened the available stocks, including RNAi lines and transposon insertions, and concluded that *Nacα* is responsible for the observed fat body cell death. Knockdown of *Nacα* in the fat body resulted in

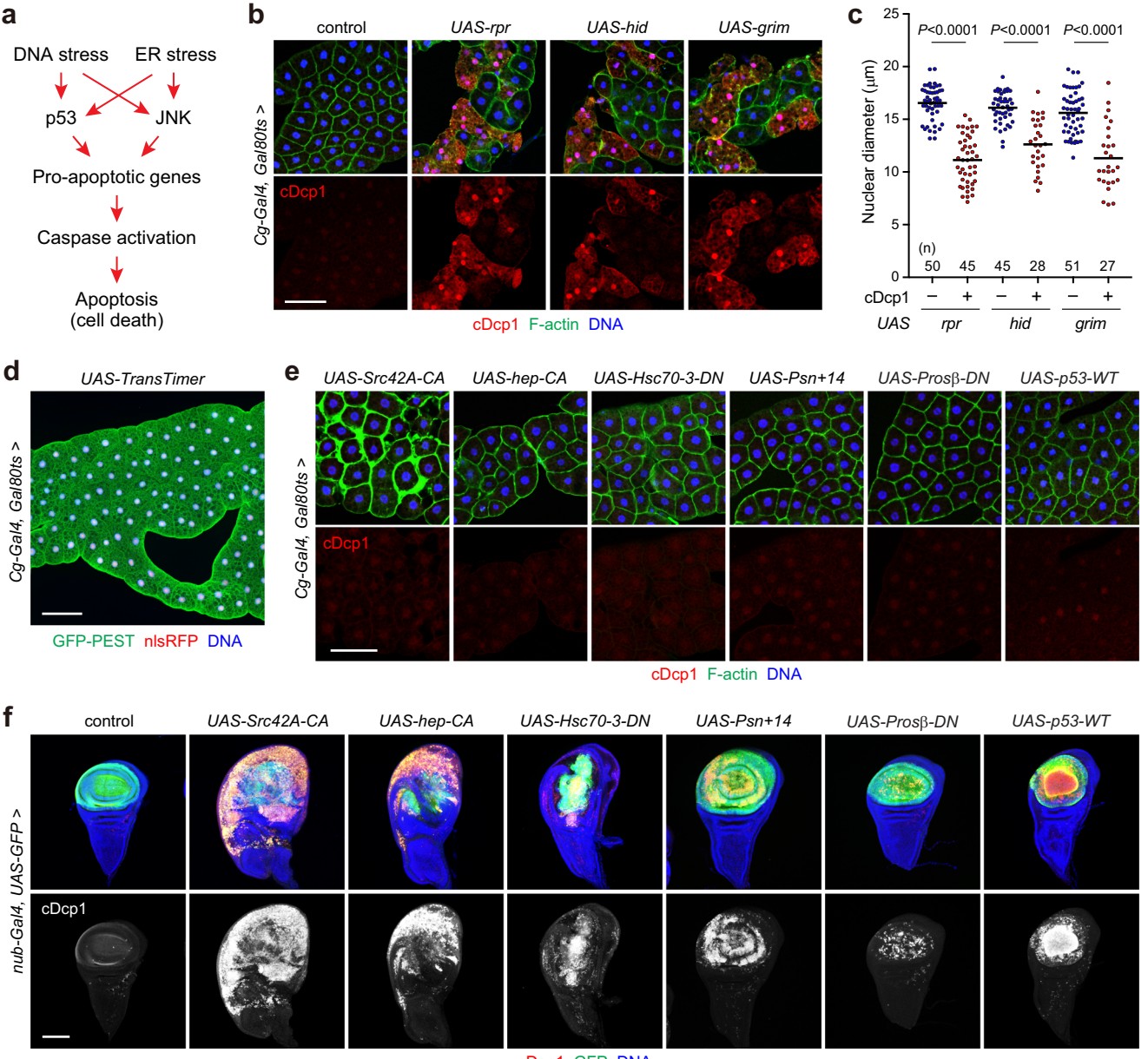

**Fig. 1 | Larval fat body is resistant to cell death by several cellular stresses.**
**a** Simplified schematic of the apoptotic signaling cascades induced by DNA and ER stresses. **b, c** Induction of cell death in the fat body by pro-apoptotic genes. Mid-third instar larvae grown at 18 °C were incubated at 29 °C for 24 h for gene induction. Fat bodies were dissected and stained for cDcp1, F-actin, and DNA (**b**). Quantification of nuclear diameter in cDcp1-negative (−) and positive (+) fat body cells in each genotype (**c**). Horizontal lines indicate the means of individual groups. Values of *n* indicate the number of nuclei from multiple animals. Unpaired two-sided Mann−Whitney *U*-test. **d** *Cg-Gal4* activity in the larval fat body by using the dual-color fluorescent reporter, TransTimer. The experimental condition is identical to those described in (**b**). Fat bodies were dissected and stained for GFP and DNA. **e** Evaluation of the cell death phenotype in the fat body. Fat bodies of the indicated genotypes were dissected from the mid-third instar larvae 24 h after the temperature shift and stained for cDcp1, F-actin, and DNA. **f** Apoptosis in wing discs. Wing discs of the indicated genotypes were dissected from the late-third instar larvae and stained for cDcp1, GFP, and DNA. *nub-Gal4* was used to drive transgene expression in the wing pouch. Scale bars: 100 μm (**b**, **d**–**f**).

massive cell death and tissue disorganization at the wandering stage (Fig. 2e and Supplementary Fig. 1e, f). Importantly, the transheterozygotes of *sug^{17Δ}* mutants over a *P*-element insertion line, *P{PZ}Nacα^{04329}* (hereafter, *Nacα^{04329}*), showed phenotypes similar to those observed in *sug^{17Δ}* homozygotes (Fig. 2c, d). Moreover, introducing a *Nacα* genomic rescue fragment (*Nacα-GR*) fully rescued the cell death phenotype in the *sug^{17Δ}* homozygote (Fig. 2d, f), indicating that the *sug^{17Δ}* allele causes dysfunction of *Nacα*.

Nacα forms a heterodimer with Nacβ, named bicaudal (bic) in *Drosophila*. The knockdown of *bic* in the fat body caused cell death (Fig. 2e and Supplementary Fig. 1f). Consistently, *bic* mutants exhibited

caspase-positive cell death in the fat body (Supplementary Fig. 1g, h). Taken together, we conclude that the Nac heterodimer is required to prevent cell death of the fat body during larval development. As our extensive attempts to segregate the *Nacα* mutant allele from *sug^{17Δ}* were unsuccessful, we hereafter refer to the original *sug^{17Δ}* mutants as *Nacα^1, sug^{17Δ}* and simply mention *Nacα^1* in this manuscript.

## A *Nacα* allele reduces its expression for cell death induction

To characterize the molecular nature of the *Nacα* mutant allele, we performed whole genome sequencing (WGS). We failed to observe any missense or nonsense mutations within the coding region (ORF) of

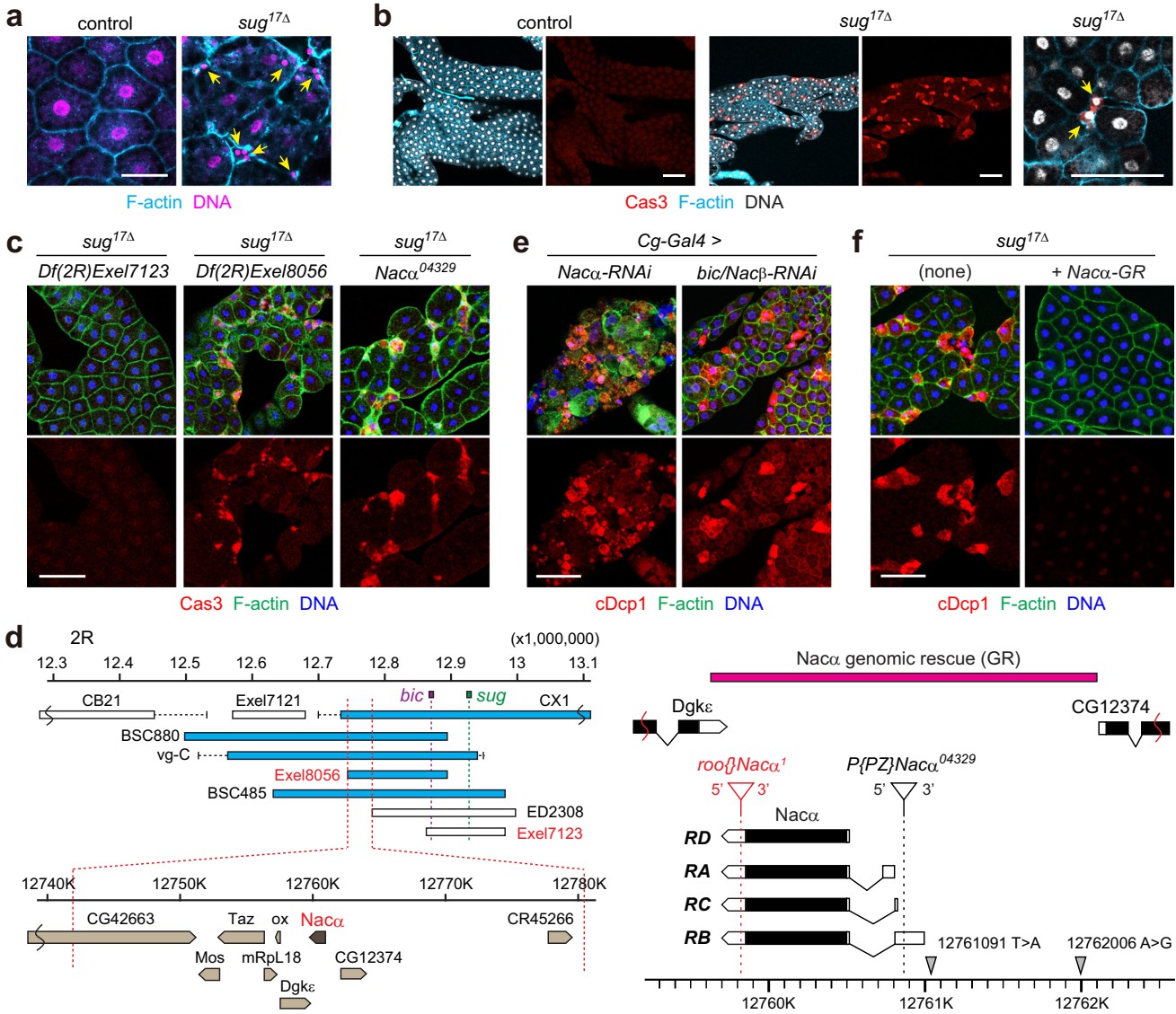

**Fig. 2 | Dysfunction of *Nacα* causes sporadic fat body cell death in *Drosophila* larvae. a** Chromatin condensation and cell shrinkage in the larval fat body. Fat bodies of the indicated genotypes were dissected from the mid-third instar and stained for F-actin and DNA. Arrows indicate small nuclei. **b** Sporadic cell death in the larval fat body. Fat bodies of the indicated genotypes were dissected from the mid-third instar and stained for Cas3, F-actin, and DNA. A representative high-magnification image is shown on the right. **c** Fat body cell death of the indicated genotypes at mid-third instar. **d** Deficiency mapping for fat body cell death. Blue bars indicate deficiencies that cause fat body cell death in *trans* with *sug¹⁷ᐃ* mutants, whereas white bars indicate deficiencies showing no cell death. The positions of the

*sug* and *bic/Nacβ* loci are indicated. Two deficiencies highlighted in red are used in **c**. A schematic representation of the *Nacα* locus is shown on the right. Protein-coding regions and untranslated regions are represented by black and white boxes, respectively. The *P*-element insertion site is marked with a white inverted triangle. A *Nacα*-GR construct is indicated by a magenta bar. The locations and nucleotide variations of the two SNPs are indicated with gray inverted triangles. **e** Knockdown of *Nacα* and *bic/Nacβ* causes cell death in the larval fat body. Fat bodies of the indicated genotypes were dissected and stained for cDcp1, F-actin, and DNA. **f** One copy of the *Nacα*-GR rescues cell death in *sug¹⁷ᐃ* homozygotes. Scale bars: 50 μm (**a**), 100 μm (**b**, **c**, **e**, **f**).

*Nacα*. There were mutations upstream of the *Nacα* transcripts (Fig. 2d and Supplementary Data 1); however, the corresponding mutations did not appear to be unique (Supplementary Fig. 2a, b). We found a break in the reads mapped to the *Nacα* 3′-UTR reference genome. A database search revealed that the adjacent sequences were identical to the long terminal repeat (LTR) of the transposable element *roo*[40]. Genomic PCR and Sanger sequencing confirmed that over 9000 bp of *roo* element was inserted at the *Nacα* 3′UTR in the inverted orientation (Fig. 2d and Supplementary Fig. 2c, d).

The retrotransposon *roo* contains an LTR at its termini and actively moves in the genome through an RNA intermediate[41]. We hypothesized that *roo* insertion would affect *Nacα* transcription in *Nacα¹* mutants. We found that *Nacα* mRNA levels were considerably

down-regulated by approximately 40% in the homozygous mutant larvae when the ORF region was amplified (Fig. 3a). Interestingly, *Nacα* transcript levels were drastically increased when a 5′UTR region was analyzed, whereas they were completely down-regulated when a 3′UTR region downstream of the *roo* insertion was analyzed (Supplementary Fig. 2e). As spliced *Nacα* transcripts are down-regulated in homozygotes, the *roo* element insertion likely prevents transcriptional progression and maturation rather than transcriptional initiation in *Nacα¹* mutants. Consistent with the results of the rescue experiments, the reduction in *Nacα* levels was recovered by introducing the *Nacα*-GR fragment (Fig. 3a). As expected, the mutant allele completely abolished the expression of *sug*. We further found that the expression of *Nacα* was reduced to approximately 20% of the control in

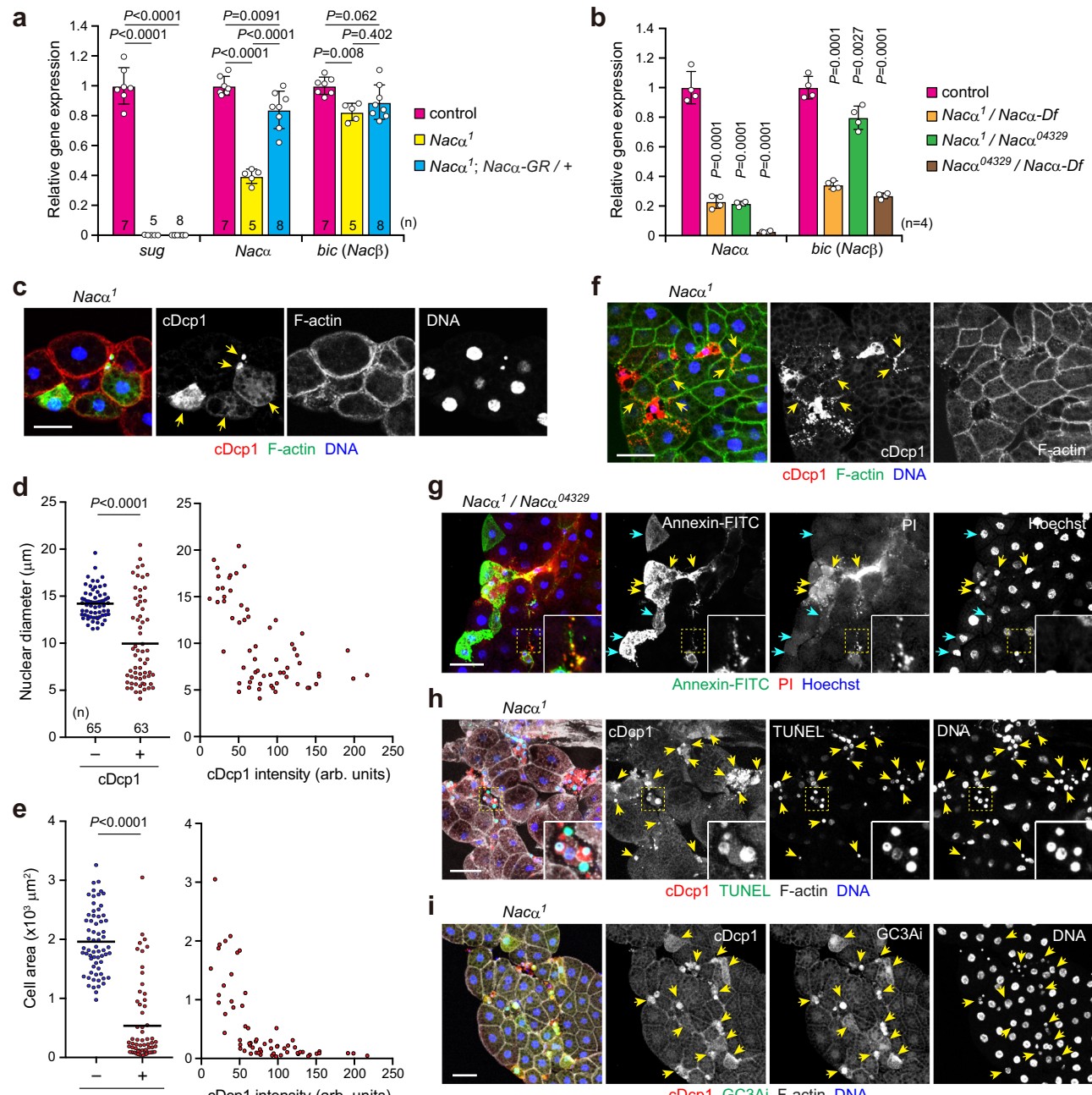

**Fig. 3 | A *Nacα* allele reduces its expression for cell death induction. a, b** Relative gene expression levels of *Nacα/β* and *sug* as determined by qRT-PCR. Second or early third instar larvae of the indicated genotypes were used. **c–e** Caspase-positive fat body cells. Fat bodies were dissected from the mid-third instar and stained for cDcp1, F-actin, and DNA (**c**). Arrows indicate cDcp1-positive cells. Quantification of nuclear diameter (**d**) and cell area (**e**) in cDcp1-negative (−) or -positive (+) fat body cells. Each plot against the cDcp1 mean fluorescence intensity per cell is shown on the right (**d, e**). **f** Formation of apoptotic bodies in *Nacα* mutant fat body. Arrows indicate cDcp1-positive dispersed puncta along the cell-cell boundaries. **g** Secondary necrosis in fat body cells. Fat bodies were dissected from the mid-

third instar and stained for Annexin V-FITC, PI, and Hoechst. Cyan and yellow arrows indicate Annexin V-positive/PI-negative and Annexin V/PI double-positive cells, respectively. **h, i** Characterization of caspase-positive fat body cells. cDcp1-positive *Nacα* mutant fat body cells are positive for TUNEL staining (**h**) and GC3Ai (**i**), as indicated by arrows. High-magnification images of dashed areas are shown (**g, h**). Horizontal lines indicate the means of individual groups (**d, e**). Values of *n* indicate the number of nuclei or cells (**d, e**) from multiple animals. For appropriate panels, results are presented as the mean ± SD (**a, b**); *n* = 4 batches (**b**); one-way ANOVA with Tukey's post hoc test (**a**), one-way ANOVA with Dunnett's post hoc test (**b**), unpaired two-sided Mann–Whitney *U*-test (**d, e**). Scale bars: 50 μm (**c, f–i**).

transheterozygotes of *Nacα¹* over a deficiency (*Df(2R)Exel8056*, hereafter referred to as *Nacα-Df*) (Figs. 2d and 3b). Similarly, the expression was reduced to 20% of the control in transheterozygotes of *Nacα¹* over *Nacα⁰⁴³²⁹* and almost completely abolished in transheterozygotes of *Nacα⁰⁴³²⁹* over *Nacα-Df*. Thus, *Nacα¹* is a weak allele, whereas *Nacα⁰⁴³²⁹* is a strong hypomorphic allele, as reported before[42].

Notably, ubiquitous knockdown of *Nacα* by *tub-Gal4* and transheterozygotes of *Nacα⁰⁴³²⁹* over *Nacα-Df* showed growth arrest and second larval lethality. Consistently, *bic* mutants, as described above, showed second larval lethality, indicating that NAC is required for normal body growth and larval development. Interestingly, the expression of *bic* was slightly reduced when *Nacα* expression was

down-regulated (Fig. 3a, b and Supplementary Fig. 1e), suggesting that *bic* responds to the expression levels of *Nacα*. As *Nacα-Df* deletes the *bic* locus (Fig. 2d), *bic* expression was reduced to more than half of the control in *Nacα¹* over *Nacα-Df* transheterozygotes (Fig. 3b).

To characterize the cell death phenotype in detail, we quantified the cDcp1 signal and nuclear size in *Nacα¹* homozygotes. The cDcp1-positive fat body cells had smaller nuclei than the surrounding cDcp1-negative cells in the mid-third instar (Fig. 3c, d). cDcp1 signals were negatively correlated with nuclear diameter. Similarly, a strong negative correlation was observed between cDcp1 signals and cell size (Fig. 3e). Dying cells often showed cDcp1- or PI-positive apoptotic bodies along the cell-cell boundaries (Fig. 3f and Supplementary Fig. 1d). Apoptotic cells expose phosphatidylserine (PS) as an eat-me signal for macrophages[43]. To further understand the fate of caspase-positive cells, we stained the cells with Annexin V, a PS-binding protein. Both Annexin V-positive/PI-negative and Annexin V/PI double-positive cells were observed in *Nacα* mutant fat bodies (Fig. 3g), which are considered early cell death and late cell death (necrotic cells), respectively. In addition, many cDcp1-positive cells were positive for the terminal deoxynucleotidyl transferase dUTP nick-end labeling (TUNEL) staining, indicating DNA fragmentation (Fig. 3h). cDcp1-positive cells were also positive for GC3Ai (Fig. 3i), a fluorescent reporter that responds to effector caspases including Dcp1 and Drice[44]. Collectively, these results suggest that caspase-positive fat body cells primarily undergo secondary necrosis, characterized by membrane breakdown and cell lysis, before phagocytosis by macrophages. While the engulfment of cell corpses has not been shown, the other experiments are consistent with apoptosis.

## Fat body cell death in *Nacα* mutants is progressive during larval development

The larval fat body consists of post-mitotic cells that undergo endoreplication[30]. The sporadic nature of fat body cell death raises questions about the timing of its onset. Therefore, we examined the cell death phenotype during the larval development. Caspase activation was observed in the early third instar, but not in the early first or early second instars (Fig. 4a). Quantitative analysis revealed that the number of smaller nuclei increased during the third instar in *Nacα¹* mutants, whereas cellular growth and endoreplication occurred normally until the third instar (Fig. 4b). Further analysis revealed that cDcp1 signals were first detected in the mid-second instar. However, nuclear and cell sizes were almost indistinguishable between cDcp1-positive and -negative neighboring cells at this stage (Supplementary Fig. 3a, b), suggesting that the cell and nuclear shrinkage/fragmentation observed in the third instar are the results of caspase activation. Similar results were observed in *Nacα¹* over *Nacα⁰⁴³²⁹* transheterozygotes (Supplementary Fig. 3c, d), suggesting that the timing of caspase activation is not related to reduced levels of *Nacα* mRNA. Consistent with this, *Nacα* knockdown fat bodies showed caspase activation and chromatin condensation/fragmentation only at the third instar stage (Fig. 4c, d), indicating that the cell death phenotype is progressive during larval development.

The accumulation of chronic cellular stress during development may trigger cell death in the third instar. Alternatively, fat body cells may have different sensitivities to *Nacα* reduction in the early and later larval stages. To distinguish between these two possibilities, we transiently knocked down *Nacα* at the mid-third instar using *Cg-Gal4* combined with *Gal80ts*. We found that cell death was induced 24 h after the temperature shift in a mosaic manner (Fig. 4e, f). Thus, the third-instar fat body is sensitive to the loss of *Nacα* and drives cell death. To further understand the cause of progressive cell death, we analyzed changes in the expression of NAC and anti-apoptotic/pro-apoptotic genes during larval fat body development. *Nacα* was almost constantly expressed during the larval stages, whereas *bic* was gradually decreased (Fig. 4g). Although caspase activation was detectable in

the second instar, the expression of the anti-apoptotic *Diap1* and pro-apoptotic genes (*rpr*, *hid*, *grim*, *Dcp1*, and *Drice*) did not noticeably change between the early second and the early third instars (Fig. 4g and Supplementary Fig. 3e). Moreover, a half-dose reduction in *Diap1* function in the *Nacα* mutant background promoted cell death in the early third instar, whereas the timing of cell death onset was not altered (Fig. 4h and Supplementary Fig. 3f, g). Thus, the timing of cell death in the fat body is likely independent of the changes in the expression of anti-apoptotic and pro-apoptotic genes.

It should be noted that the nuclei of some *Nacα* knockdown cells became abnormally larger than those of control cells (Fig. 4d). The chromosomes in these cells were less densely packed (Supplementary Fig. 3h), suggesting that the nucleus underwent expansion rather than endoreplication. Moreover, these cells were either negative or weakly positive for cDcp1 staining. Thus, the persistent pro-apoptotic status of undead cells that fail to undergo cell death causes nuclear expansion for unknown reasons. Most *Nacα* knockdown larvae were eventually arrested at the wandering stage and failed to pupariate.

## The larval fat body is a sensitive tissue that leads to cell death by *Nacα* reduction

In several animal species, the loss of NAC causes cellular defects and cell death in a cell type- and stage-specific manner[45–48]. Thus, we investigated the tissue specificity of the cell death phenotype in *Nacα* mutant larvae. To this end, we analyzed cDcp1 immunoreactivity in several tissues during the wandering stage. *Nacα¹* homozygotes displayed increased cDcp1 signals only in the fat body, whereas other tissues, such as salivary glands, ring glands, imaginal discs, and the central nervous system (CNS), were indistinguishable from control larvae (Fig. 5a, b). In contrast, *Nacα¹* over *Nacα⁰⁴³²⁹* transheterozygotes showed increased cDcp1 signals in the fat body, imaginal discs, and the corpora allata of the ring gland, an endocrine organ that generates juvenile hormones[49].

Given that *Nacα¹* mutants have reduced expression, tissue specificity may arise from the different *Nacα* expression levels; therefore, we analyzed *Nacα* expression in each tissue by quantitative reverse transcription PCR (qRT-PCR). *Nacα¹* homozygotes showed decreased mRNA levels (<50% of the control) in the fat body, salivary glands, imaginal discs, and central brain (Supplementary Fig. 4a). *Nacα¹* over *Nacα⁰⁴³²⁹* transheterozygotes showed approximately half the expression levels of *Nacα¹* homozygotes, further validating the *Nacα* expression levels in *Nacα¹* mutants. These results suggest that each tissue has a different threshold for *Nacα* reduction, which triggers cell death. Alternatively, *Nacα* may not always be dispensable, depending on the cell type. To clarify these two possibilities, we focused on the salivary glands, the best-studied polyploid cells in larvae. *Nacα* knockdown in the salivary glands using *fkh-Gal4* caused cell death and tissue disorganization at the wandering stage (Supplementary Fig. 4b), indicating that *Nacα* is indispensable for cell survival in the salivary glands. Together, these results suggest that the fat body is a sensitive tissue that activates the cell death program by *Nacα* reduction, compared to other tissues, including the imaginal discs and salivary glands.

## Loss of *Nacα* induces ER stress and cell death through JNK signaling

Next, we examined the signaling pathways that induce fat body cell death in *Nacα* mutants. In several species and cell lines, the loss of NAC activates the ER stress response, which eventually leads to cell death through JNK and caspase activation[21,46,47]. To evaluate ER stress in vivo, we utilized the ER stress reporter Xbp1-EGFP, in which GFP is expressed only when ER stress triggers the splicing of *Xbp1* mRNA[16]. Knockdown of *Nacα* increased nuclear GFP signals in the early third instar larvae, indicating that the loss of Nacα activates ER stress in the fat body (Fig. 6a, b). Consistently, the *Nacα¹* mutant fat body increased spliced *Xbp1* (*Xbp1s*) mRNA levels, compared with unspliced *Xbp1* (*Xbp1u*)

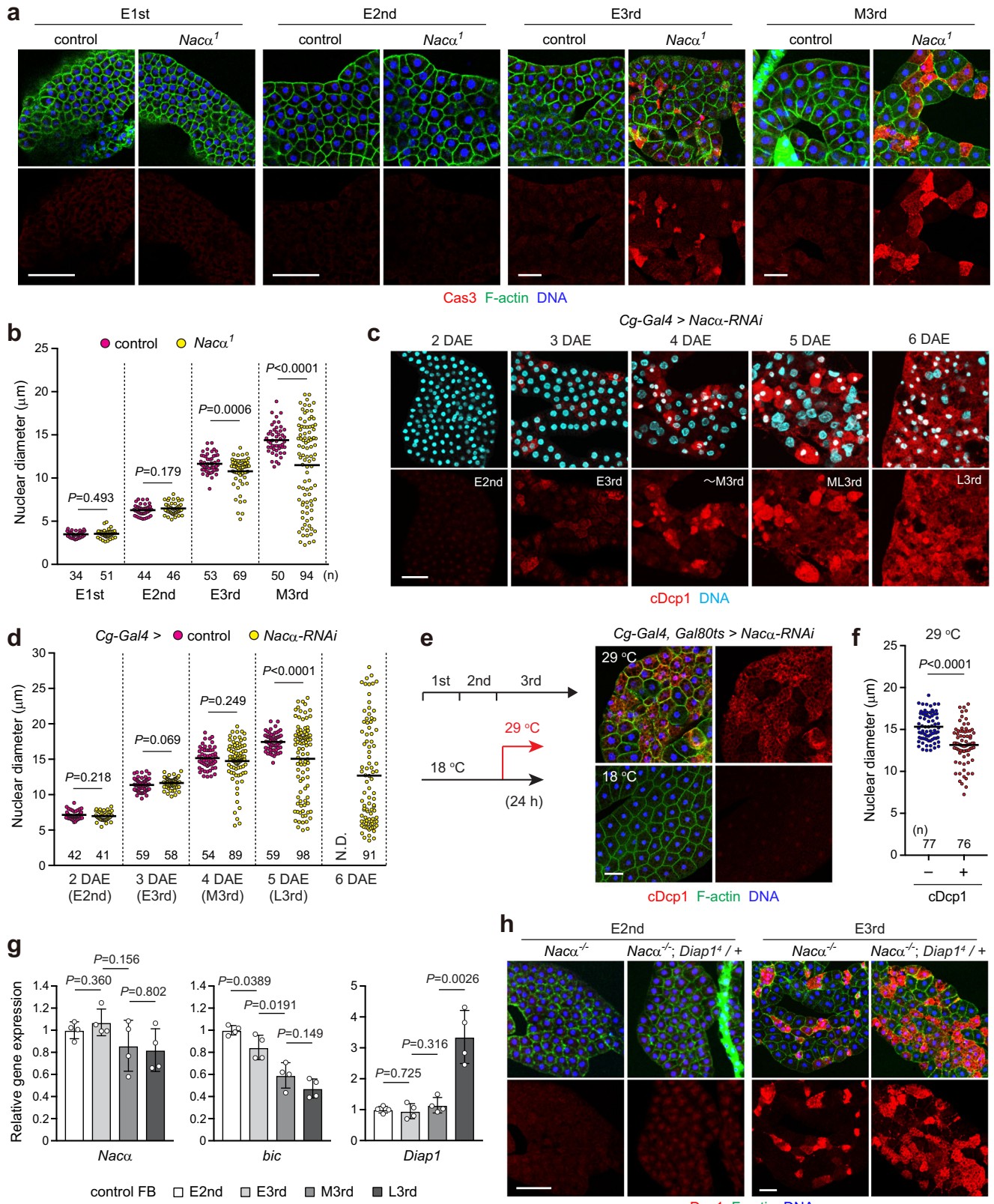

(Fig. 6c). The expression of Xbp1-target genes, such as *Hsc70-3* and *Protein disulfide isomerase* (*Pdi*), was also increased in *Nacα¹* mutants. Moreover, *Nacα¹* mutants showed increased expression of the known JNK target gene *puckered* (*puc*). The induction of these genes was completely reversed by the presence of the *Nacα*-GR fragment, indicating that the induction of ER stress and JNK activation are due to the

dysfunction of *Nacα* and not the deletion of *sug*. Knockdown of *bic* also increased Xbp1-EGFP and Xbp1-/JNK target gene expression in the fat body (Supplementary Fig. 5a–c), further indicating that the NAC complex functions to prevent ER stress.

Previous studies have suggested that chronic ER stress induces JNK-dependent and -independent apoptosis in *Drosophila* imaginal

**Fig. 4 | Fat body cell death in *Nacα* mutants is progressive during larval development. a, b** Fat body cell death in *Nacα* mutants during larval development. Fat bodies of the indicated genotypes were dissected from each stage and stained for Cas3, F-actin, and DNA (**a**). Quantification of the nuclear diameter in fat body cells (**b**). E1st, early first instar; E2nd, early second instar; E3rd, early third instar; M3rd, mid-third instar (24 h after E3rd). **c, d** Fat body cell death in *Nacα* knockdown larvae. Fat bodies were dissected daily and stained for cDcp1, F-actin, and DNA (**c**). As the *Nacα* knockdown larvae delayed the development, average larval stages of the knockdown animals are indicated. DAE, days after eclosion; ML3rd, mid-late third instar before the wandering stage; L3rd, late third instar. Quantification of nuclear diameter in fat body cells (**d**). Stages in the parentheses indicate larval stages in control animals. ND, not determined due to puparium formation. **e, f** Transient knockdown of *Nacα* at the third instar is sufficient to induce fat body cell death. The experimental scheme for the temperature shift is shown on the left.

Early third instar larvae grown at 18 °C were collected and maintained at either 18 °C or 29 °C for 24 h. Fat bodies were dissected and stained for cDcp1, F-actin, and DNA (**e**). Quantification of the nuclear diameter in cDcp1-negative (−) or -positive (+) fat body cells (**f**). **g** Relative gene expression levels of *Nacα/β* and *Diap1*. Fat bodies (FB) of the control larvae were dissected from the indicated stages and subjected to qRT-PCR analysis. **h** Genetic interaction of *Diap1* with *Nacα*. Fat bodies of the indicated genotypes were dissected from each stage and stained. *Nacα* $^{-/-}$, *Nacα$^1$/ Nacα$^{04329}$* mutants. Quantification of the nuclear diameter and cDcp1 signals is shown in Supplementary Fig. 3f, g. Horizontal lines indicate the means of individual groups (**b, d, f**). Values of *n* indicate the number of nuclei (**b, d, f**) from multiple animals. For appropriate panels, results are presented as the mean ± SD, *n* = 4 batches (**g**); unpaired two-tailed Welch's *t*-test (**b, d, f**); unpaired two-tailed Student's *t*-test (**g**). Scale bars, 50 μm (**a, c, e, h**).

discs[34,36]. Therefore, we investigated the involvement of JNK signaling in fat-body cell death. The expression of a dominant-negative *bsk* (*Drosophila* JNK) entirely suppressed cDcp1 staining induced by *Nacα* knockdown (Fig. 6d), suggesting that caspase activation occurs downstream of the JNK signaling pathway. Expression of the baculovirus anti-apoptotic gene *p35* completely suppressed cDcp1 immunoreactivity. Importantly, inhibition of the JNK pathway and caspase activity suppressed cellular shrinkage and chromatin condensation (Fig. 6e), suggesting that cell and nuclear shrinkage are part of the cell death process mediated by JNK. However, disorganization of F-actin staining was still observed in *Nacα* knockdown fat bodies when cell death was blocked (Fig. 6d). To further examine the consequences of *Nacα* reduction, we analyzed the cell death phenotype in wing discs. The knockdown of *Nacα* by *nub-Gal4* increased cDcp1-positive cells in wing discs (Fig. 6f and Supplementary Fig. 5d) and led to tiny wings in adults (Fig. 6g). Co-expression of *bsk-DN* and *p35* fully suppressed *Nacα* knockdown-induced cell death at the wandering stage. However, the small wing phenotype in adults was not rescued by inhibition of the JNK pathway or caspases. It is possible that residual caspase activity causes the wing phenotype during metamorphosis. These results suggest that dysfunction of proteostasis by loss of *Nacα* disrupts normal wing development.

ER stress evokes a series of cellular defense mechanisms, such as the activation of autophagy and the lysosomal degradation pathway, which resolve protein aggregates[17]. We found that the knockdown of *Nacα* did not induce LysoTracker-positive autolysosomes in the fat body at the early third instar (Supplementary Fig. 5e), suggesting that dysfunction of Nacα does not activate autophagy despite increased ER stress at this stage (Fig. 6a, b). Short-term starvation increased LysoTracker-positive signals in both control and *Nacα* knockdown fat bodies. Vacuole-like autolysosomes were eventually observed in *Nacα* knockdown fat bodies, indicating that abnormal autolysosomes were formed coincident with cell death. However, a dominant-negative *Atg1* (*Atg1-KQ*) did not affect caspase activation induced by *Nacα* knockdown (Fig. 6d, e). Similar results were observed in wing discs (Fig. 6f and Supplementary Fig. 5d), suggesting that *Nacα* knockdown-mediated cell death is independent of autophagic activity, and thus not autophagic cell death.

In imaginal discs, activated JNK signaling up-regulates a series of target genes for tissue repair and compensatory proliferation, in addition to inducing apoptosis[50,51]. JNK also regulates the transcriptional activator yorkie (yki), which is inhibited by the Hippo pathway. In addition to the induction of *Xbp1s* and *puc*, we found that *Nacα* knockdown in fat bodies drastically increased the expression of known JNK target genes, including matrix metalloproteinases (*Mmp1* and *Mmp2*), cytokine-like ligands (*upd1*, *upd2*, and *upd3*), and a relaxin-like peptide hormone *dilp8* (Fig. 6h and Supplementary Fig. 5f). Moreover, some yki target genes are up-regulated, such as *CycE* and *Diap1*. In addition,

several mitogens, including *wg*, *dpp*, and *hh*, were up-regulated. These results suggest that larval fat bodies respond similarly to imaginal discs when JNK signaling is activated.

## *Nacα* mutants impair cytosolic proteostasis and increase p53 proteins for cell death induction

Previous studies demonstrated that the p53 gene is expressed in the larval fat body, but its protein is not detectable because of the proteasomal degradation system[31]. We found that the p53 protein was detectable by western blotting (WB) in *Nacα* knockdown, but not in the control fat body (Fig. 7a). Consistently, nuclear localization of the protein-fusion fluorescent reporter p53-GFP was detected in cDcp-1-positive fat body cells (Fig. 7b), suggesting that *Nacα* mutants showed increased p53 protein levels. Notably, some, but not all, cDcp1-negative cells showed nuclear signals of p53-GFP in both *Nacα* mutant and knockdown fat bodies (Fig. 7b and Supplementary Fig. 6a), suggesting an instructive role of p53 in cell death induction. Furthermore, *p53* mRNA levels were dramatically increased in *Nacα* or *bic* knockdown fat bodies (Fig. 7c and Supplementary Fig. 5f). Consistently, the expression levels of pro-apoptotic genes (*rpr*, *hid*, and *grim*) were noticeably increased. Therefore, we examined the contribution of p53 in fat body cell death. The expression of a dominant-negative *p53* (*p53-DN*) partially suppressed the increased cDcp1 staining in *Nacα* mutant fat bodies (Fig. 7d and Supplementary Fig. 6b), indicating that p53 is involved in the observed cell death. Given that inhibition of the JNK pathway by *bsk-DN* almost entirely suppressed cell death in *Nacα* mutants, the JNK pathway appears to play a dominant role in caspase activation.

In addition to transcriptional induction, the up-regulation of endogenous p53 proteins in the larval fat body implies that *Nacα* mutants impair proteasomal degradation. This hypothesis is supported by the fact that *Nacα* mutant worms increase the cytosolic stress response, presumably due to the accumulation of misfolded ER and secreted proteins in the cytosol by mistargeting and the ERAD system[18,52]. To examine this possibility, we analyzed proteasome activity using the fluorescence probe Me4BodipyFL-Ahx3Leu3VS, which covalently binds to the active proteasome. Although the fluorescence signals varied to some extent between cells in the control fat body, *Nacα* mutants showed lowered signals (Fig. 7e, f). Co-incubation with the proteasome inhibitor, MG132, appreciably decreased the observed signals, suggesting probe specificity in the larval fat body. Given that a few cells undergo cell death in *Nacα* mutant fat body, *Nacα* mutants seem to reduce proteasome activity globally. Furthermore, fluctuations in the number of overexpressed p53 proteins between cells further supported the intrinsic heterogeneity of proteasome activity in the control larvae (Supplementary Fig. 6c–e). Consistent with the fluorescence probe results, the MG132-sensitive peptidase activity of the 20S/26S proteasome was considerably decreased in *Nacα* mutant fat body cell lysates (Fig. 7g). Moreover, *Nacα* mutants and *Nacα/bic* knockdown increased ubiquitinated proteins in the fat

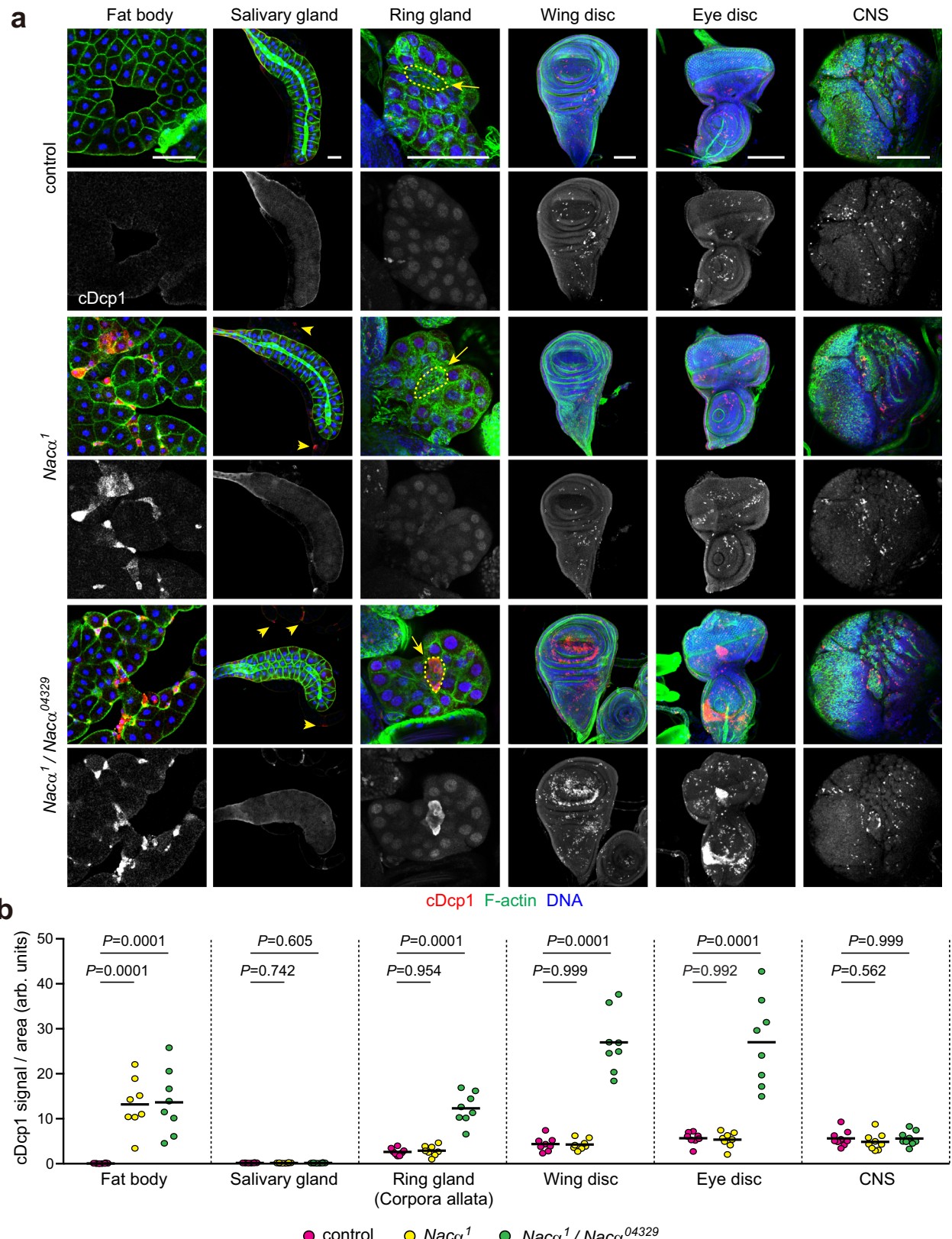

**Fig. 5 | Larval fat body is a sensitive tissue that leads to cell death by *Nacα* reduction. a, b** Cell death in *Nacα* mutant tissues. Each tissue of the indicated genotypes was dissected from the late third instar and stained for cDcp1, F-actin, and DNA (**a**). Arrowheads indicate cDcp1-positive fat body cells. Arrows and dashed areas indicate the regions of the corpora allata. CNS, central nervous system. Quantification of the cDcp1 mean fluorescence intensity per tissue area (**b**). *n* = 8 (fat body, salivary glands, ring glands, wing discs, and eye discs) or 9 (CNS) tissues from multiple animals; one-way ANOVA with Dunnett's post hoc test. Scale bars, 100 µm.

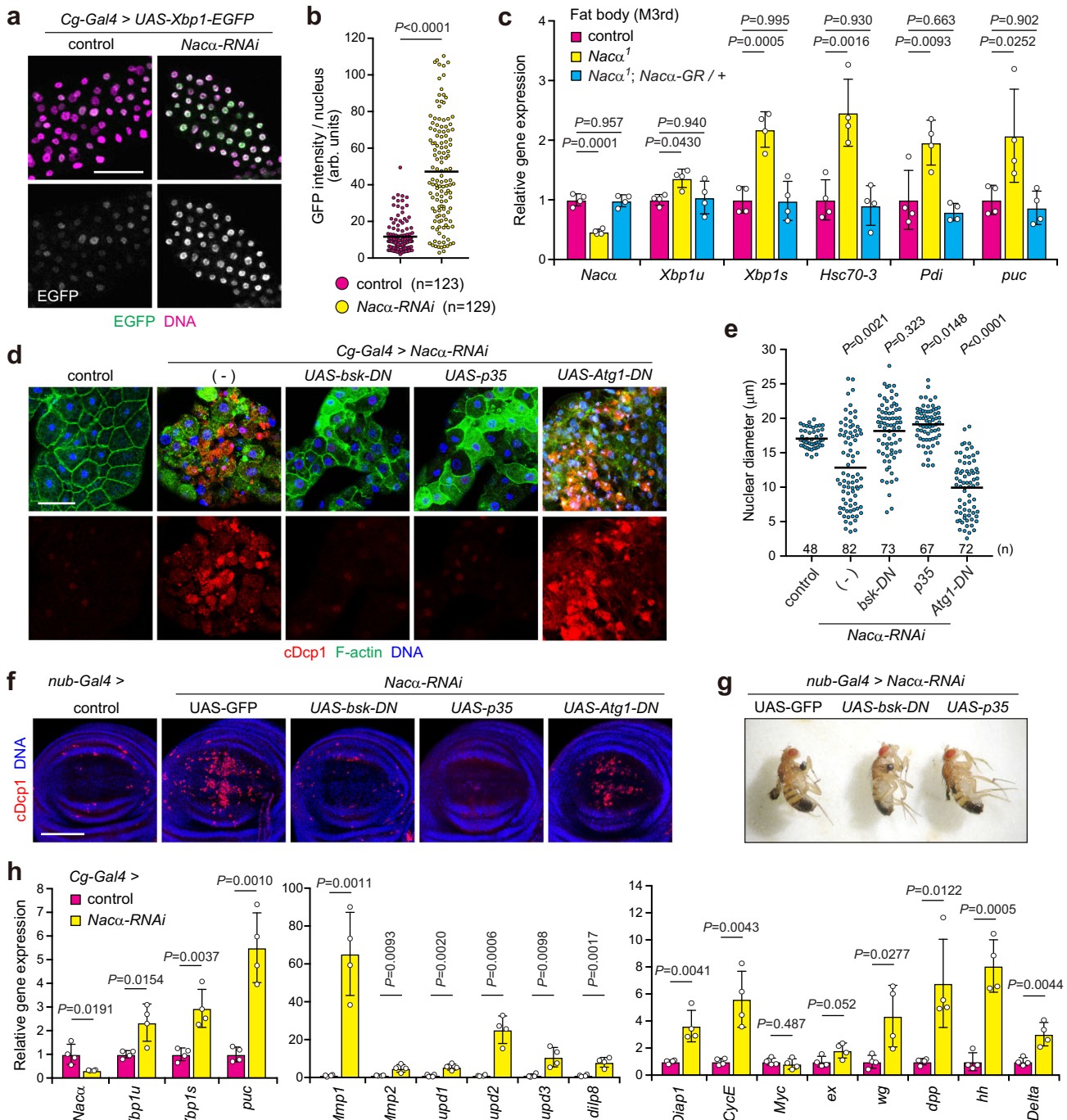

**Fig. 6 | Loss of *Nacα* induces ER stress and cell death through JNK signaling.**
**a**, **b** ER stress as determined by the expression of Xbp1-GFP. Fat bodies of the indicated genotypes were dissected from the early third instar and stained for GFP and DNA (**a**). Quantification of the mean GFP fluorescence intensity per nucleus (**b**). **c** Relative gene expression levels in fat bodies as determined by qRT-PCR. Fat bodies of the indicated genotypes were dissected from the mid-third instar (M3rd) and subjected to the experiments. **d**, **e** Cell death in the fat body. Fat bodies of the indicated genotypes were dissected from the late third instar (L3rd) and stained for cDcp1, F-actin, and DNA (**d**). Quantification of the nuclear diameter of fat body cells (**e**). **f** Apoptosis in wing discs. Wing discs of the indicated genotypes were dissected

from the L3rd and stained for cDcp1 and DNA. The quantification of cDcp1 signals is shown in Supplementary Fig. 5d. **g** Adult wing morphology of the indicated genotypes. Representative images of male flies are shown. **h** Relative gene expression levels in *Nacα* knockdown fat body. Fat bodies were dissected from the M3rd and subjected to qRT-PCR analysis. Horizontal lines indicate the means of individual groups (**b**, **e**). Values of *n* indicate the number of cells (**b**) or nuclei (**e**) from multiple animals. For appropriate panels, results are presented as the mean ± SD, *n* = 4 batches (**c**, **h**); unpaired two-sided Mann–Whitney *U*-test (**b**), one-way ANOVA with Dunnett's post hoc test (**c**), Kruskal–Wallis test followed by Dunn's post hoc test (**e**), unpaired two-tailed Student's *t*-test (**h**). Scale bars, 50 μm (**a**), 100 μm (**d**, **f**).

body (Fig. 7h and Supplementary Fig. 6f–j), suggesting that reducing the NAC function impairs the ubiquitin-mediated proteasomal degradation system.

Low proteasome activity evokes a cytosolic stress response called the HSR[6,7]. Oxidative stress also facilitates protein misfolding and aggregation, thereby inducing the HSR. Moreover, ROS mediates the

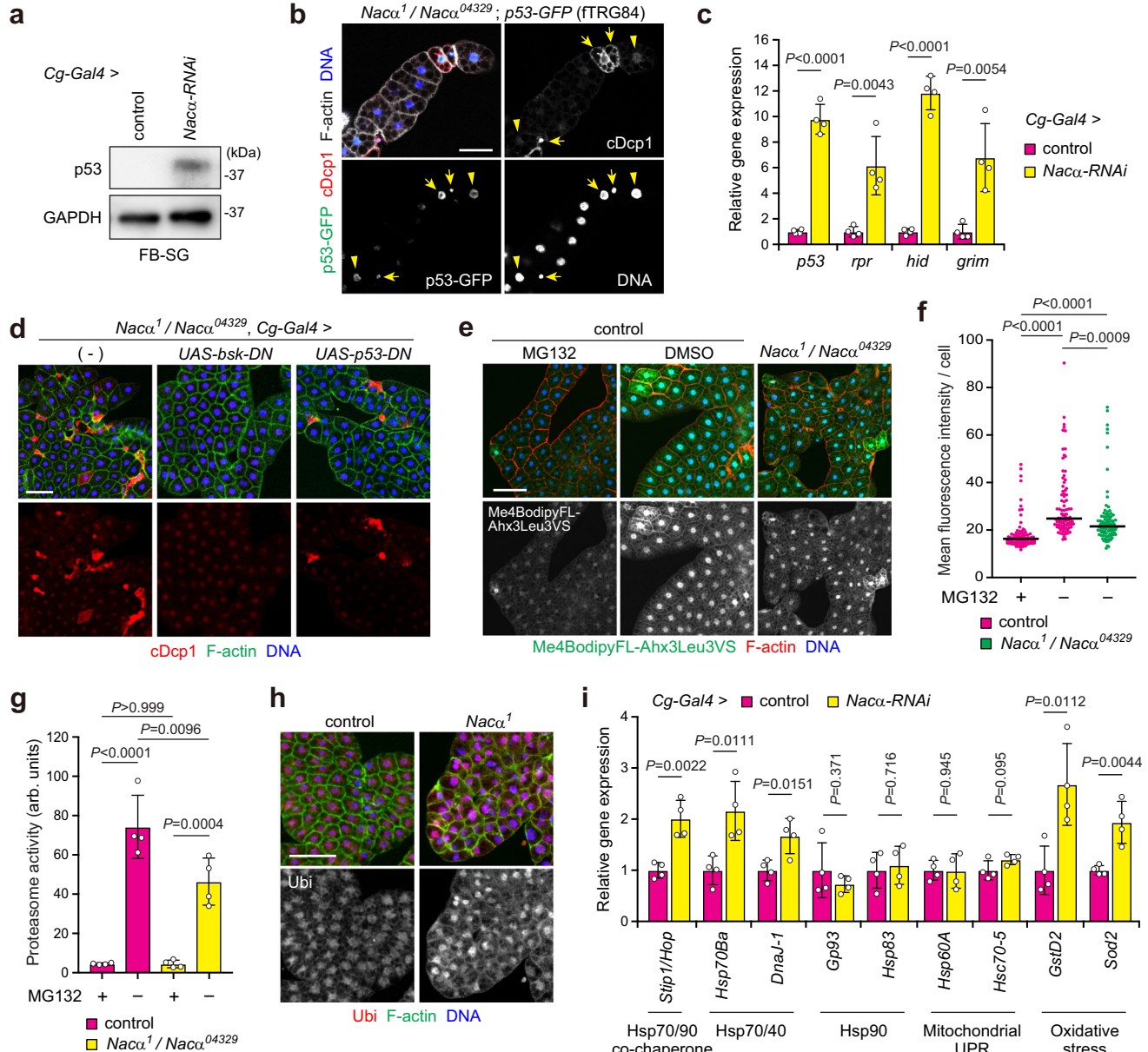

**Fig. 7 | Nacα mutants impair cytosolic proteostasis and increase p53 proteins for cell death induction. a** Amount of p53 protein in the fat body. Fat bodies of the indicated genotypes were dissected from the mid-third instar (M3rd) and used for western blot analysis. **b** p53 protein levels in Nacα mutant fat body cells. Fat bodies of the indicated genotypes were dissected from the M3rd and stained for GFP, cDcp1, F-actin, and DNA. Arrows and arrowheads indicate p53/cDcp1 double-positive and p53-positive/cDcp1-negative cells, respectively. **c** Relative gene expression levels in Nacα knockdown fat body. Fat bodies were dissected from the M3rd and used for qRT-PCR analysis. **d** Cell death in the fat body. Fat bodies of the indicated genotypes were analyzed at the M3rd. The quantification of cDcp1 signals is shown in Supplementary Fig. 6b. **e, f** Proteasome activity in the fat body as determined using a fluorescence probe. Fat bodies of the indicated genotypes were dissected from the M3rd, incubated with Me4BodipyFL-Ahx3Leu3VS, and then stained to detect F-actin and DNA (**e**). Quantification of Me4BodipyFL-Ahx3Leu3VS mean fluorescence intensity per cell (**f**). **g** Proteasome activity in the fat body. Peptidase activity was measured using a synthetic substrate in the fat body cell lysate, dissected from the M3rd. **h** Ubiquitin levels in the fat body. Fat bodies of the indicated genotypes were dissected from the early third instar and stained for ubiquitin, F-actin, and DNA. The quantification of ubiquitin signals is shown in Supplementary Fig. 6j. **i** Relative gene expression levels in Nacα knockdown fat body cells. Fat bodies were dissected from the M3rd and used for qRT-PCR analysis. Horizontal lines indicate the medians of individual groups, $n = 90$ cells from multiple animals (**f**). For appropriate panels, results are presented as the mean ± SD, $n = 4$ batches (**c, g, i**); unpaired two-tailed Student's $t$-test (**c, i**), Kruskal–Wallis test followed by Dunn's post hoc test (**f**), one-way ANOVA with Tukey's post hoc test (**g**). Scale bars, 50 μm (**b**), 100 μm (**d, e, h**).

crosstalk between p53 and JNK in several contexts[50,51]. We found that Nacα knockdown increased HSR, as indicated by the up-regulation of Hsp70/40 (Hsp70Ba and DnaJ-1) and its co-chaperone Stip1 mRNA levels (Fig. 7i and Supplementary Fig. 5f). The expression of oxidative stress genes (GstD2 and Sod2) also increased. In contrast, the expression of Hsp90 genes (Gp93 and Hsp83) and mitochondrial UPR genes (Hsp60A and Hsc70-5) did not change. Taken together, these results

suggest that Nacα reduction impairs both cytosolic and ER proteostasis.

**Reduction of Nacα sensitizes the fat body to pro-apoptotic and environmental stresses**
Next, we genetically examined the contribution of defective proteostasis and oxidative stress to fat body cell death in Nacα mutants.

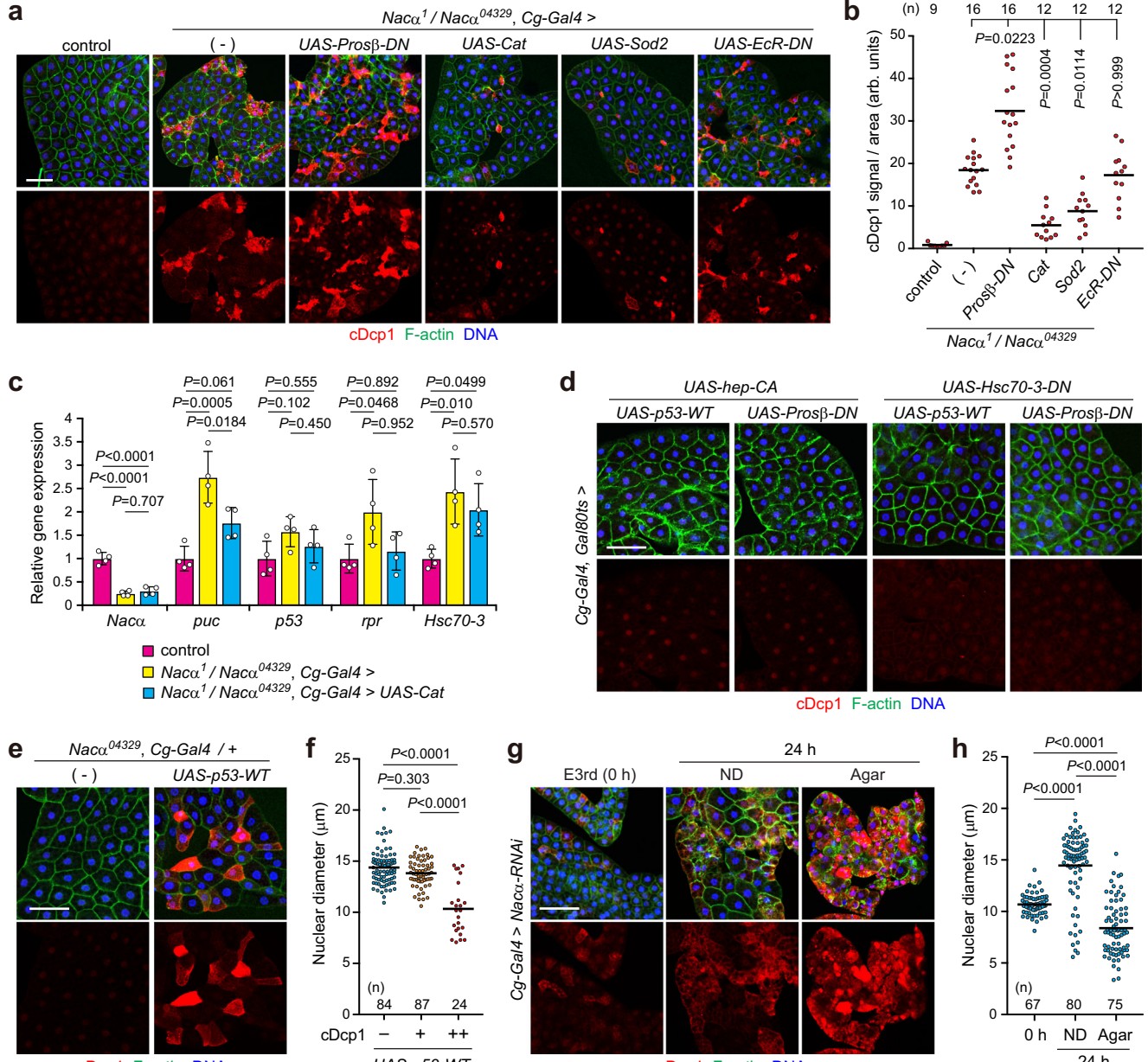

**Fig. 8 | Reduction of *Nacα* sensitizes the fat body to pro-apoptotic and environmental stresses. a, b** Cell death in the fat body. Fat bodies of the indicated genotypes were dissected from the mid-third instar (M3rd) and stained for cDcp1, F-actin, and DNA (**a**). Quantification of the mean cDcp1 fluorescence intensity per tissue area (**b**). **c** Relative gene expression levels in *Nacα* mutant fat body cells. Fat bodies of the indicated genotypes were dissected from the M3rd and used for qRT-PCR analysis. **d** Evaluation of cell death phenotype in the fat body. Fat bodies of the indicated genotypes were dissected from the M3rd 24 h after the temperature shift and stained for cDcp1, F-actin, and DNA. **e, f** Overexpression of p53 induces fat body cell death in *Nacα* heterozygous mutants. Fat bodies of the indicated genotypes were dissected from the M3rd and stained for cDcp1, F-actin, and DNA (**e**).

Quantification of nuclear diameter in cDcp1-negative (−), -positive (+), or strongly positive (++) fat body cells (**f**). **g, h** Starvation accelerates cell death in the *Nacα* knockdown fat body. Early third instar larvae (E3rd) were reared on a normal diet (ND) or fasted on an agar-only diet (agar) conditions for 24 h. Fat bodies were dissected at the indicated time points and stained (**g**). Quantification of the nuclear diameter of fat body cells (**h**). Horizontal lines indicate the means of individual groups (**b, f, h**). Values of *n* indicate the number of tissue area (**b**) or nuclei (**f, h**) from multiple animals. For appropriate panels, results are presented as the mean ± SD, *n* = 4 batches (**c**); Kruskal–Wallis test followed by Dunn's post hoc test (**b, f, h**), one-way ANOVA with Tukey's post hoc test (**c**). Scale bars, 100 μm (**a, d, e, g**).

Inhibition of proteasomal activity by *Prosβ-DN* markedly increased the number of cDcp1-positive cells in *Nacα* mutant background (Fig. 8a, b). This suggests that the proteasomal degradation system protects fat body cells from caspase activation in *Nacα* mutants. Moreover, forced expression of the ROS scavenger enzymes, *Catalase* (*Cat*) and *Superoxide dismutase 2* (*Sod2*), partially alleviated the cell death phenotype in *Nacα* mutants, suggesting the involvement of ROS in cell death induction. Consistent with the cell death phenotype, *Cat* expression

partially suppressed *puc* and *rpr* expression in *Nacα* mutants (Fig. 8c). Notably, *Cat* expression had little effect on the increased ER chaperone gene expression, suggesting that ROS mainly act downstream of proteotoxic stress to boost p53 and JNK activation. We also inhibited the action of ecdysteroids, in which hormone levels sharply increased during the third instar[53]. However, a dominant-negative *EcR* (*EcR-DN*) had no effect on the cell death phenotype (Fig. 8a, b). Moreover, feeding *Nacα* knockdown larvae with 20-hydroxyecdysone did not

promote cell death induction (Supplementary Fig. 7a–c), indicating that ecdysteroid signaling is not critically involved in fat body cell death in *Nacα* mutants.

The JNK and p53 pathways are major pro-apoptotic factors in *Drosophila* and play prominent roles in inducing cell death. Our results demonstrate that *Nacα* mutants cause fat body cell death dependent on both the JNK and p53 pathways, although the activation of each pathway was insufficient for cell death induction (Fig. 1e). Therefore, we tested whether combining ER stress (or JNK activation) and impaired cytosolic proteostasis (or the p53 pathway) induces cell death. However, co-expression of *hep-CA* together with *p53-WT or Prosβ-DN* failed to induce cell death (Fig. 8d and Supplementary Fig. 7d). The expression of *p53-WT or Prosβ-DN* with increased ER stress showed no phenotype. Thus, although the JNK and p53 pathways are involved in fat body cell death in *Nacα* mutants, additional factors may contribute to maintaining the adaptive response by preventing cell death.

To further demonstrate the physiological importance of Nacα, we examined whether *Nacα* reduction renders fat body cells susceptible to cell death due to intrinsic and environmental stresses. To test this hypothesis, we overexpressed p53 in an *Nacα* heterozygous mutant background. Forced expression of *p53* induced caspase activation in *Nacα04329* heterozygous fat body cells (Fig. 8e, f). Thus, the loss of one copy of *Nacα* sensitizes fat body cells to p53-induced cell death. The fat body stores nutritional reserves and plays a prominent role in generating energy sources during starvation[54,55]. In the control larvae, starvation did not induce cell death in the fat body (Supplementary Fig. 7e). However, 24 h of starvation in the early third instar drastically accelerated cell death in *Nacα* knockdown larvae compared with those fed a normal diet (Fig. 8g, h). We further found that 12 h of starvation considerably increased ER stress in *Nacα* knockdown larvae, although starvation decreased ER stress in control larvae (Supplementary Fig. 7f, g). Consistently, the expression of *bsk-DN* completely suppressed the observed caspase activation in *Nacα* mutants (Supplementary Fig. 7h), suggesting that JNK signaling plays an essential role in starvation-induced cell death. These results further suggest that Nacα is crucial for protecting the fat body from cell death induced by pro-apoptotic and environmental stresses.

### Proteostasis stress is the leading cause of caspase activation and cell death induction in *Nacα* mutants

Finally, we examined whether enhanced protein-folding suppresses fat body cell death. For this, we used the chemical chaperone, 4-phenylbutyrate (4-PBA), which prevents the accumulation of unfolded proteins in the ER[47,56]. Feeding larvae with 4-PBA partially suppressed starvation-induced cell death (Fig. 9a–c), supporting the hypothesis that ER stress is a cause of cell death. We also reduced translation initiation with rapamycin, a potent and specific inhibitor of mTor. Rapamycin strongly suppressed starvation-induced cell death, as evaluated by cDcp1 staining and nuclear condensation/fragmentation. Consistently, the expression of a dominant-negative form of mTor (*mTor-DN*) largely suppressed cell death in *Nacα* knockdown larvae (Fig. 9d, e). Inhibition of mTor activity dramatically reduced the nuclear and cell size in the fat body (Supplementary Fig. 8a, b), suggesting reduced protein translation.

To directly investigate whether reducing overall protein synthesis attenuates the cell death phenotype, we focused on *Minute* mutants, a series of heterozygous mutants of the ribosomal protein gene[57,58]. *Minute* mutants show a considerable delay in their larval period, but adult flies are essentially normal. Recent structural analysis revealed that Nacβ binds to RpL22 and RpL19, whereas Nacα does not bind directly to the ribosome[52]. Because no *Minute* mutants have been reported for the *RpL22* gene[59], we used two *Minute* mutants, *RpL19* and *RpS26*. Heterozygous *Minute* mutants (*RpL19/+* and *RpS26/+*) did not induce cell death in the fat body (Fig. 9f), whereas a half-dose

reduction in ribosome function caused apoptosis in developing wing discs (Supplementary Fig. 8c, d), as reported previously[60–62]. Under these conditions, *Minute* mutants partially suppressed fat body cell death caused by Nacα dysfunction (Fig. 9g, h). In contrast, the increased apoptosis in *Nacα* mutant wing discs was not altered or even exacerbated in the heterozygous *Minute* mutant background (Supplementary Fig. 8c, d). These results suggest that *Nacα* genetically interacts with ribosomal protein genes in a tissue-dependent manner.

Interestingly, the expression of the ER chaperones, *Hsc70-3* and *Pdi*, increased during development, peaking at the mid-third instar (Supplementary Fig. 3e), suggesting an increased demand for protein folding in the fat body. Collectively, these results strongly suggest that proteotoxic stress is the leading cause of caspase activation and cell death induction in *Nacα* mutant fat body (Fig. 9i).

## Discussion

In this study, we showed that proteostasis is a crucial determinant of cell survival or death in a cell type-specific manner. We identified a new *Nacα* allele that induces sporadic cell death in the larval fat body. The insertion of a *roo* transposable element at the *Nacα* 3′UTR is likely responsible for the reduced *Nacα* expression and *Nacα1* mutant phenotype shown in this manuscript. The *sug174* mutant was generated by CRISPR/Cas9-mediated deletion of the coding region[39]. Recent studies have indicated that double-strand breaks induced by CRISPR/Cas9 facilitate the transposition of autonomously active retrotransposons in mammals[63]. *Roo* is the most abundant retrotransposon in the *Drosophila* genome, some of which retain transpositional activity[64,65]. Given the low sequence homology between the guide RNA and the insertion site of the *roo* element, it remains unclear whether transposition in the *Nacα* locus occurs spontaneously or by the CRISPR/Cas9-system. Notably, the original study analyzed *sug174* mutants in *trans* with a deficiency allele (*Df(2R)Exel7123*) that does not disrupt the *Nacα* locus in all experiments[39].

Polyploid cells, including the larval fat body, are protected from p53-dependent cell death by at least two mechanisms: p53 protein degradation and transcriptional silencing of p53-target loci[31]. We attempted to reconstruct cell death by genetic manipulation of ER stress and cytosolic proteostasis, which are the primary targets affected by Nacα dysfunction; however, none of these tools induced cell death, even when combined. Given that ectopic induction of pro-apoptotic genes induces cell death in the fat body, ER and proteotoxic stresses are likely insufficient to overcome the transcriptional silencing of pro-apoptotic genes. This idea is strongly supported by the fact that overexpressed p53 induces cell death in the salivary glands when epigenetic regulators are knocked down simultaneously[31]. Although the crosstalk between the Nacα/β heterodimer and epigenetic regulation remains unclear, a recent report revealed that the knockdown of *Nacα* causes misexpression of the Hox gene *Abd-B* during heart development[48]. *bic* was originally identified as a mutant that affects the anterior/posterior axis of the embryo by regulating *oskar* mRNA localization[42,66]. Moreover, NACα regulates the transactivation of AP-1 target genes in mammals by binding to the c-Jun proto-oncogene[67,68]. Thus, Nacα dysfunction may directly or indirectly affect multiple cellular events, including epigenetic states, transcription, and translation, ultimately resulting in caspase-dependent cell death in the fat body.

One prominent feature of fat body cell death observed in *Nacα* mutants and by pro-apoptotic genes is mosaicism within tissues. Interestingly, cell death processes, including cellular/nuclear shrinkage and secondary necrosis, occur slowly following caspase activation. Although larval fat body cells are considered to have equal cell identity, our recent histochemical analysis revealed that metabolic states prominently differ between cells[69,70]. Moreover, we found that ER stress and proteasome activity fluctuated between cells in the control larvae. A similar mosaicism has been reported in *rpr-lacZ* expression when *p53* is overexpressed in the salivary glands[26]. The function of

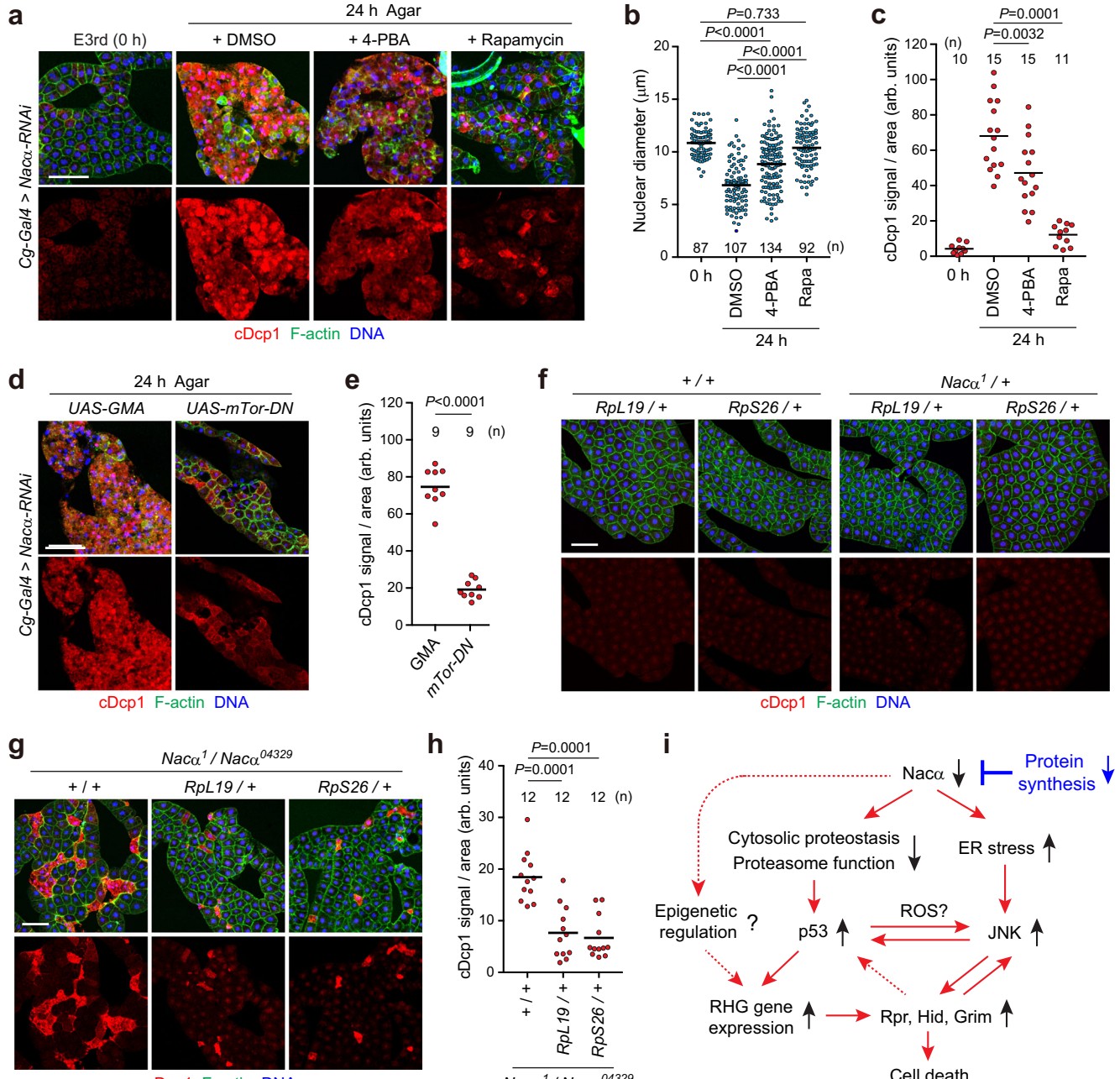

**Fig. 9 | Proteostasis stress is the leading cause of caspase activation and cell death induction in *Nacα* mutants. a–c** Cell death in the *Nacα* knockdown fat body is partially alleviated by 4-PBA and rapamycin. Early third instar larvae (E3rd) were fasted for 24 h on an agar-only diet (agar) supplemented with 4-PBA or rapamycin. Fat bodies were dissected at the indicated time points and stained for cDcp1, F-actin, and DNA (**a**). Quantification of the nuclear diameter (**b**) and mean cDcp1 fluorescence intensity per tissue area (**c**) is shown. **d, e** Inhibition of mTor suppresses fat body cell death in *Nacα* knockdown larvae. E3rd larvae of the indicated genotypes were fasted for 24 h, fat bodies were dissected at the indicated time points, and stained (**d**). Quantification of the mean cDcp1 fluorescence intensity per tissue area (**e**). *UAS-GMA*, GFP-tagged actin-binding domain of *Drosophila* moesin,

was used as a control. **f** Evaluation of the cell death phenotype in the fat body. Fat bodies of the indicated genotypes were dissected from the M3rd and stained for cDcp1, F-actin, and DNA. **g, h** One copy loss of ribosomal protein genes in *Nacα* mutant background. Fat bodies of the indicated genotypes were dissected from the M3rd and stained for cDcp1, F-actin, and DNA (**g**). Quantification of the mean cDcp1 fluorescence intensity per tissue area (**h**) is shown. **i** Proposed model of fat body cell death in *Nacα* mutants. RHG genes, *rpr*, *hid*, and *grim*. Horizontal lines indicate the means of individual groups (**b, c, e, h**). Values of *n* indicate the number of nuclei (**b**) or tissue area (**c, e, h**) from multiple animals. Kruskal–Wallis test followed by Dunn's post hoc test (**b**), one-way ANOVA with Dunnett's post hoc test (**c, h**), unpaired two-sided Mann–Whitney *U*-test (**e**). Scale bars, 100 μm (**a, d, f, g**).

histone demethylases in opening chromatin silencing requires a metabolic intermediate, α-ketoglutarate; thus, epigenetic modification is tightly coupled to its cellular metabolism[71]. Owing to the nature of polyploidy in the fat body and salivary glands, subtle differences in epigenetic and metabolic states between cells may contribute to cell fates that undergo cell death. Consistent with this, starvation stress

may accelerate epigenetic changes that result in caspase activation in *Nacα* mutants. It has been reported that positive feedback loops between p53 and JNK amplify caspase signals for cell death in wing discs[72]. Pro-apoptotic genes function downstream and upstream of JNK signaling to boost caspase-dependent cell death. Based on these observations, it would be intriguing to hypothesize that cell death is

induced in fat body cells, which eventually amplifies the feedback loops between p53 and JNK in weakened metabolic/epigenetic states. This idea further explains why cell death progresses gradually for over half a day in the fat body.

We also demonstrated that the fat body is a sensitive tissue that undergoes cell death by reducing *Nacα* mRNA levels. Furthermore, fat bodies in early larval stages are relatively resistant to cell death. The fat body secretes a vast majority of proteins, including extracellular matrix (ECM) and larval serum proteins (Lsps)[73,74]. Previous studies have suggested that highly secretory cells exhibit elevated ER stress during normal development[16]. Interestingly, *Nacα* mutants in zebrafish cause the specific disappearance of mesenchymal stromal cells by apoptosis during the larval stages[47]. In mammals, mesenchymal cells are sensitive to ER stress-induced apoptosis, owing to the high secretory activity of ECMs and matricellular proteins[75,76]. As endoreplication of fat body cells enhances protein production capacity during the larval stages, the initiation of cell death in *Nacα* mutants is likely related to the increased protein synthesis activity. However, ER stress in the fat body is relatively weaker than that in other secretory cells, such as the salivary gland and midgut[15,16]. Thus, additional factors may be involved in the fat body-specific cell death phenomena in *Nacα* mutants. A plausible explanation is its unique metabolic activity, such as de novo lipogenesis, as it occurs in hepatocytes[77]. Accumulation of lipid intermediates and saturated fatty acids is associated with lipotoxicity and potentially elevates ER stress by altering ER integrity and ROS production[78]. Moreover, the fat body expresses cytochrome p450 monooxygenases that catalyze detoxification and produce ROS[30]. Thus, higher mitochondrial activity and ROS production may be related to tissue specificity of the cell death phenotype. Although further analysis is required to clarify the molecular mechanisms underlying increased ER stress upon starvation in *Nacα* mutants, starvation-induced cell death may also be attributed to incomplete lipid catabolism and lipotoxicity in the fat body.

We showed that inhibition of mTor and ribosomal protein deficiency alleviated the cell death phenotype in the *Nacα* mutant fat body. In wing discs, ribosomal protein deficiency induces proteotoxic stress and JNK activation, resulting in apoptosis[60–62]. However, the stoichiometric imbalance of ribosome proteins does not induce cell death in the fat body, consistent with proteasome disruption and JNK activation not inducing cell death in this tissue. Thus, we speculate that reducing NAC function is likely to cause more deleterious cellular stress in the fat body than that caused by ribosomal protein deficiency. Given that ribosome deficiency protects against ER stress in yeast[79], the beneficial effects of ribosome deficiency outweigh any toxic effects, resulting in the prevention of cell death in the *Nacα* mutant background. Although there is an obvious difference in cell-cycle state, the larval fat body upregulates several mitogens and cytokines by *Nacα* knockdown, similar to damaged imaginal discs[50,51]. It remains unclear whether the induction of these molecules is a physiological response that serves as a compensatory signal toward neighboring fat body cells or is simply a result of abnormal JNK signaling coinciding with cell death. Given that the fat body recognizes the nutritional state and remotely regulates body growth[80,81], it would be interesting to explore the impact of sporadic cell death and/or reduction in fat body cell number on systemic body growth and metabolism.

In mammals, some polyploid cells, such as the non-proliferating trophoblast giant cells, are resistant to DNA damage and cell death[27]. In addition, the polyploid state plays a tumor-suppressive role by mutagens in the liver[82]. Importantly, cancer cells often have complex karyotypes, and some cells have elevated genomic content and become polyploid tumor cells[83,84]. Polyploidy facilitates tumor evolution and the acquisition of apoptosis-resistant characteristics in multiple incurable cancers, although polyploidy does not necessarily drive resistance to cell death. Thus, the *Nacα* mutants in *Drosophila* not only

advance our understanding of larval fat bodies, but also open new avenues for future research and strategies for cancer therapy.

## Methods

### *Drosophila* strains

The following *Drosophila melanogaster* strains were used: *w[1118]* (used as a control), *sug[A17]* (a gift from Dr. Ville Hietakangas), *UAS-bsk-DN, nub-Gal4* (gifts from Dr. SaKan Yoo), *UAS-Atg1-DN* (a gift from Dr. Thomas P. Neufeld), and *UAS-grim* (a gift from Dr. Michael B. O'Connor). *Nacα-RNAi* (8759R-1), *bic-RNAi* (3644R-1), and *Diap1-RNAi* (HMS00752) were obtained from the National Institute of Genetics (NIG) *Drosophila* Stock Center. *P{lacW}RpL19[k03704]* (102285), *Diap1[4]* (108011), *P{lacW} Diap1[j5C8]* (111396), and *P{SUPor-P}RpS26[KG00230]* (114620) were obtained from the Kyoto *Drosophila* Genetic Resource Center (DGRC). *Nacα-RNAi* (GD, 36017), *Nacα-RNAi* (KK, 109114), *bic-RNAi* (KK, 104718), and p53-GFP (fTRG84, 318453) were obtained from the Vienna *Drosophila* RNAi Center (VDRC). *Nacα-RNAi* (NIG) line was used for the analysis, except for Supplementary Fig. 1f. *bic-RNAi* (KK) line was used for the analysis shown in Supplementary Fig. 5a–c, f. *bic-RNAi* (NIG) line was used for the analysis shown in Supplementary Fig. 6h, i. The following stocks were obtained from the Bloomington *Drosophila* stock center (BDSC): *UAS-p35* (5073), *Tub-Gal4* (5138), *UAS-rpr* (5824), *UAS-Hsc70-3-D231S* (5841), *UAS-hep-CA* (6406), *UAS-Src42A-CA* (6410), *UAS-p53-259H* (6582), *UAS-p53-WT* (6584), *UAS-Prosβ6[1]*, *UAS-Prosβ2[1]* (6787), *Cg-Gal4* (7011), *UAS-mTor.TED* (7013), *Df(2R)Exel7123* (7870), *Df(2R)Exel8O56* (7916), *UAS-EcRA-W650A* (9451), *P{lacW}bic[K10712]* (10998), *P{PZ}Nacα[04329]* (11371), *P{SUPor-P}bic[KG01035]* (13186), *UAS-Sod2* (24494), *UAS-Cat* (24621), *UAS-GMA* (31775), *UAS-Xbp1-EGFP* (60730), *UAS-psn + 14* (63243), *UAS-hid* (65403), *fkh-Gal4* (78061), *UAS-GC3Ai* (84343), and *UAS-TransTimer* (93411).

### Fly food and developmental staging

The animals were reared on fly food (normal diet, ND) that contained 8 g agar, 100 g glucose, 45 g dry yeast, 40 g corn flour, 4 ml propionic acid, and 0.45 g butylparaben (in ethanol) per liter. Yeast paste was not added to the fly tubes in any of the experiments. All the experiments were conducted under non-crowded conditions at 25 °C, unless otherwise indicated. For the transient starvation experiments, staged larvae were washed in phosphate-buffered saline (PBS) to remove all traces of food and transferred to a vial containing an agar-only diet for defined time periods. For feeding experiments, 4-phenylbutyric acid (Tokyo Chemical Industry), rapamycin (Tokyo Chemical Industry), and 20-hydroxyecdysone (Sigma) were added to the fly food or agar-only diet at a final concentration of 15 mM, 50 μM, and 0.1 mg/g diet, respectively. Developmental staging was performed as previously described[70,85]. Adult flies were photographed under a Zeiss Stemi 2000-C stereomicroscope (Zeiss) equipped with a Canon PowerShot G15 digital camera (Canon).

### Deficiency screening

As the *sug[A17]* homozygotes were semi-lethal whereas *sug[A17]* over *sug-Df* was viable, we reasoned that a second site mutation related to fat body cell death may be linked to lethality. Thus, for the first screening, developmental lethality of *sug[A17]* mutants was analyzed in *trans* with the following deficiency lines obtained from the DGRC: DrosDel Core Deletion Kit for second chromosome modified by DGRC Kyoto, 2L (48 lines) and 2R (29 lines), and selected deficiency strains for second chromosome, Deficiency kit 2 (97 lines). For the second screening, deficiency lines that showed lethality in *trans* with *sug[A17]* mutants were analyzed for fat body cell death at the mid-third instar by PI and anti-Cas3 staining, as described below.

### WGS and mapping

Genomic DNA was extracted from ten homozygous *sug[A17]* mutant adults using a DNeasy Blood & Tissue kit (#69504, QIAGEN). The eluted

samples were further extracted using phenol-chloroform and purified through ethanol precipitation. The DNA samples were checked and visualized on a 0.8% agarose gel run in 0.5× TAE buffer to ensure the presence of high molecular weight DNA. The DNA extracts were quantified using a NanoDrop ND-1000 spectrophotometer (Thermo Fisher Scientific). A DNA library was prepared using the TruSeq DNA PCR-free kit (Illumina) and sequenced with 150 bp paired-end reads on an Illumina instrument in a commercial facility (Macrogen Corp. Japan).

The obtained reads were mapped onto the reference genome sequence (BDGP Release 6) using CLC Genomics Workbench software (version 11.0.1, QIAGEN). Single nucleotide polymorphisms (SNPs) and insertions/deletions (indels) were identified by comparing mutant reads with reference sequences. Detected SNPs and indels were further checked for specificity using *Drosophila* Genetic Reference Panel 2 (DGRP2) inbred lines derived from a natural population[86]. A detailed summary of SNPs and indels is provided in Supplementary Data 1. SNPs near the *Nacα* locus were manually analyzed using a BLAST search against the nucleotide collection. Sequence alignments, shown in Supplementary Fig. 2, were performed using the ClustalW program (https://www.genome.jp/tools-bin/clustalw). Reads were deposited in the Sequence Read Archive (SRA) data repository of the National Center for Biotechnology Information (NCBI) as Bioproject PRJNA941141 [https://www.ncbi.nlm.nih.gov/sra/SRP425742].

### Plasmid construction
The DNA fragment of the *Nacα* genomic region was cloned by genomic PCR using sequenced iso-1 strain (#2057, BDSC) and subcloned into the pCR-BluntII-TOPO vector (Invitrogen). The fragment was subcloned into pCaSpeR4 vector, and then transformants were obtained using a standard injection method (WellGenetics). We analyzed two independent transformants inserted into the second and third chromosomes, respectively, and obtained the same results.

### Quantitative reverse transcription PCR (qRT-PCR) analysis
qRT-PCR analysis was performed as previously described[87,88]. Total RNA was prepared using TRIzol (Invitrogen) or the RNeasy mini kit with RNase-Free DNase Set (QIAGEN), and reverse transcription was performed using the PrimeScript RT Master Mix (Takara Bio). qRT-PCR was performed on an ABI PRISM 7500 (Thermo Fisher Scientific) using TB Green Premix Ex TaqII (Takara Bio). Transcript levels of intended mRNA were normalized with *rp49* levels in the same samples. Primers used in this study are listed in Supplementary Data 2.

### Immunohistochemistry and image quantification
Larval tissues were dissected in PBS, fixed for 10 min in 3.7% formaldehyde in PBS containing 0.2% TritonX-100, and processed as previously described[70,85]. The following primary and secondary antibodies were used: rabbit anti-cleaved Caspase-3 (1/200, #9661, Cell Signaling Technology), rabbit anti-cleaved Dcp-1 (1/200, #9578, Cell Signaling Technology), mouse anti-ubiquitin (1/200, P4D1, sc-8017, Santa Cruz), mouse anti-LacZ (1/200, 40-1a, Developmental Studies Hybridoma Bank), chicken anti-GFP (1/200, ab13970, Abcam), Alexa Fluor 488-conjugated goat anti-chicken IgY (1/500, A-11039, Thermo Fisher Scientific), Alexa Fluor 488-conjugated goat anti-mouse IgG (1/500, A-11029, Thermo Fisher Scientific), Alexa Fluor 555-conjugated goat anti-rabbit IgG antibodies (1/500, A-21428, Thermo Fisher Scientific), and Alexa Fluor 555-conjugated goat anti-mouse IgG (1/500, A-21424, Thermo Fisher Scientific). The F-actin was stained with Alexa Fluor 488-Phalloidin (1/500, A12379, Thermo Fisher Scientific), Alexa Fluor 555-Phalloidin (1/500, A34055, Thermo Fisher Scientific), or Alexa Fluor 647-Phalloidin (1/500, A22287, Thermo Fisher Scientific). The nuclei were stained with Hoechst 33342 (H3570, Thermo Fisher Scientific), DAPI (D523, Dojindo), or PI solution (P378, Dojindo).

For the proteasome activity probe, fat bodies were dissected in PBS and incubated in Schneider's *Drosophila* medium (21720024, Thermo Fisher Scientific) containing 1 μM Me4BodipyFL-Ahx3Leu3VS (I-190, R&D Systems) with or without 2 μM MG132 (135-18453, Wako) for 30 min at room temperature before fixation. For staining of acidic organelles including autolysosome, larval tissues were dissected in PBS, incubated with 100 nM LysoTracker Red (L7528, Thermo Fisher Scientific) in PBS containing Hoechst 33342 for 2 min at room temperature, washed twice in PBS, and mounted in 50% glycerol in PBS. TUNEL assay and Annexin V staining were performed using ApopTag® Red In Situ Apoptosis Detection Kit (S7165, Merck) and Annexin V-FITC kit (#4700, Medical & Biological Laboratories), respectively, according to the manufacturer's instructions.

Images were acquired with a Zeiss LSM700 confocal laser scanning microscope with Zen 2009 software or LSM800 microscope with Zen 2.3 software. The obtained images were processed and analyzed using Fiji software (version 2.3.0, NIH). For the quantification of cDcp1 signals per cell, as shown in Fig. 3d and e, the mean fluorescence intensity in each cell was manually analyzed based on F-actin signals. The nuclear diameter and cell size were manually measured based on DNA and F-actin signals, respectively. For quantification of Xbp1-GFP signals, the nuclear area was traced based on the DNA signals, and the GFP mean fluorescence intensity was analyzed. To quantify cDcp1 signals per tissue area, the tissue area was traced based on DNA and F-actin signals, and cDcp1 mean fluorescence intensity was analyzed. To quantify the p53 and ubiquitin signals shown in Supplementary Fig. 6e, i, j, the nuclear area was automatically traced based on the DNA signals, and the mean fluorescence intensities were analyzed.

### Proteasome peptidase activity
Larval fat bodies were dissected in PBS and homogenized in a buffer containing 25 mM Tris-HCl (pH 7.5), 2 mM ATP, 5 mM MgCl$_2$, and 1 mM dithiothreitol. Proteasome peptidase activity in the lysates was measured with 1 μM of a synthetic peptide substrate, succinyl-Leu-Leu-Val-Tyr-7-amino-4-methyl-coumarin (Suc-LLVY-AMC) (S-280-05M, R&D Systems), with or without 2 μM MG132 (135-18453, Wako) at room temperature.

### WB analysis
Larval tissues were dissected in PBS and homogenized in 100 μl 1x SDS sample buffer using a pellet pestle and boiled for 10 min. The samples were centrifuged at 15,000 rpm (20,000 × g) for 5 min at room temperature and the supernatants were subjected to sodium dodecyl-sulfate polyacrylamide gel electrophoresis (SDS-PAGE) followed by WB using a poly(vinylidene fluoride) membrane (Immobion-P, Millipore). Primary and secondary antibodies used were mouse monoclonal anti-p53 (1/100, C-11, sc-55476, Santa Cruz), mouse monoclonal anti-ubiquitin (1/100, P4D1, sc-8017, Santa Cruz), goat polyclonal anti-GAPDH (1/500, IMG-3073, Imgenex), HRP-linked anti-mouse IgG (1/2000, #7076, Cell Signaling Technology), and HRP-linked anti-goat IgG (1/2000, ab97110, Abcam). The proteins were visualized using ECL Prime western blotting detection reagent (Amersham) and detected by a LuminoGraph II imaging system (ATTO). Images were processed and analyzed by Fiji.

### Statistics and reproducibility
All experiments were repeated at least twice using independently reared populations to ensure reproducibility. Alternatively, samples were collected from populations that were independently reared. For image quantification, tissues, and cells were randomly selected from multiple samples derived from several animals and analyzed using the Fiji. Representative images obtained from several animals are presented. The experiments were not randomized, and the investigators were not blinded to the fly genotypes during the experiments. The sample sizes were chosen based on the number of independent

experiments required for statistical significance and technical feasibility. The sample size for each experiment and the stages of larvae used for the analysis are indicated in the figures and figure legends. Statistical tests were performed using Microsoft Excel version 16.54 or GraphPad Prism version 7.0. The statistical tests used are described in the figure legends.

### Reporting summary

Further information on research design is available in the Nature Portfolio Reporting Summary linked to this article.

## Data availability

The source data underlying all main and supplementary figures are provided as a Source Data file. Sequences of oligonucleotides used in this study are included in Supplementary Data 2. WGS data are deposited in the SRA under accession number PRJNA941141. Source data are provided with this paper.

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

## Acknowledgements

We thank Ville Hietakangas, Thomas P. Neufeld, Hiroshi Nishida, Michael B. O'Connor, SaKan Yoo, the Bloomington Stock Center (BDSC), the Vienna *Drosophila* RNAi Center (VDRC), the Kyoto *Drosophila* Genetic Resource Center (DGRC), and the National Institute of Genetics (NIG) *Drosophila* Stock Center for fly stocks and the Developmental Studies Hybridoma Bank for antibodies; SaKan Yoo for fruitful discussions and critical comments on the manuscript; Naoki Okamoto for comments on the manuscript. This work was supported by the Japan Society for the Promotion of Science (JSPS) [KAKENHI grants 21H02495], the Takeda Science Foundation, and the RIKEN BDR-Otsuka Pharmaceutical Collaboration Center (RBOC) funding program Kakehashi to T.N.

## Author contributions

T.Y. and T.N. designed the experiments. T.Y., Y.Y., H.O. and T.N. performed the experiments. T.Y., M.T. and T.N. performed the data analysis. T.N. supervised the work and wrote the manuscript. All authors discussed and reviewed the manuscript.

## Competing interests

The authors declare no competing interests.
