## [Peer Review File · Nature Communications]

REVIEWER COMMENTS

Reviewer #1 (Remarks to the Author):

In this manuscript, the authors report that Nascent polypeptide-associated complex protein alpha subunit (NACa) is essential for protein homeostasis in *Drosophila* fat body cells. Specifically, the authors use independent genetic conditions (different alleles and RNAi lines) to show that even a mild loss of NACa activates cell death in the fat body. Stronger loss of function alleles interfere with cell survival in other tissues. They further show that Naca deficient fat body cells activate various stress response pathways, including p53, JNK, ER stress response, ROS response.

As the authors cite, Nac had been established as a critical regulator of cellular homeostasis. Along these lines, a recent *Drosophila* study (not cited by the authors) showed that Naca RNAi interferes with (heart) tissue development (Schroeder et al., PLoS Genetics PMID 36240221). The current manuscript does a fine job of describing the phenotypes of Naca loss. But given our existing knowledge of Nac, the overall message here appears somewhat of an expected outcome. The induction of apoptosis, in particular, involves the usual suspects such as p53, JNK, and RHG genes. Could the authors provide some previously unknown conceptual insights into the standard phenotypic description? How about adding new mechanistic insights regarding why fat body cells are more vulnerable to a mild loss of Naca? Alternatively, how about demonstrating whether ER stress (or possibly other sources of stress) is the leading “cause” of cell death induction in the fat body?

Technically, the authors have done an excellent job with proper controls and quantification of data. Below are points that could help enhance the conceptual depth of this manuscript for the authors' consideration:

1. It would be helpful to the readers if the authors introduced Nac in the Introduction. Specifically, a brief summary of what is already known about the Naca in cellular homeostasis and what are the critically unanswered questions would be helpful. As it is, the background information is sporadically introduced in the Results section, and the outstanding questions about the Nac complex are not well spelled out.
2. The authors use the term “apoptosis” loosely throughout the manuscript. That term is based on morphological features that include cell-corpse engulfment. Since most fat body cells do not undergo engulfment, it is probably better to use the term “caspase-induced cell death.”
3. In Figure 6, the authors detect UPR signaling markers (such as spliced Xbp1) after Naca loss. Based on this, the authors imply that apoptosis is induced by ER stress (line 291, 292). However, there is no experiment here establishing “causality” of ER stress. Moreover, the authors show in Figure 7I that

cytoplasmic chaperones are also induced. It remains unclear whether the ER stress markers merely correlate with cell death, or indicate causality.

4. One way to enhance the depth of the study is to demonstrate what is the cause of cell death induction in Naca mutants. Can the authors suppress the fat body cell death phenotype by enhancing protein misfolding in the ER or the cytoplasm? Or alternatively, can they suppress the phenotype by reducing overall protein synthesis (thereby reducing defective nascent peptides)?

5. Figure 6G: p35 does not fully suppress the wing phenotype. The authors conclude that Naca loss disrupts normal wing development irrespective of cell death. This conclusion could be misleading because p35 may not fully block apoptosis (I point out that the authors have not examined apoptosis markers in pupal stages). Also, p35 does not block initiator caspases such as Dronc (Meier et al. EMBO J. PMID 10675329), and therefore, it remains possible that the residual Dronc activity is causing the wing phenotype.

6. The authors demonstrate “causality” of ROS in Figure 8A by overexpressing ROS quenchers such as Cat or Sod2. This is excellent. Could they use this assay to demonstrate that ROS is a cause of p53 and JNK induction?

7. What happens if one overexpresses NACa in the fat body. Would that protect those cells from adverse conditions?

Reviewer #2 (Remarks to the Author):

Yamada et al start by observing that the sug17 mutant allele has a second site mutation that causes small nuclei/apoptosis in the fat body. They pursue this to discover a hypomorphic mutation in the Naca gene, encoding part of the heterodimer that binds to the nascent ribosome-associated polypeptides, caused by a roo insertion in the 3UTR of Naca, causing a reduction in the transcription of Naca. I find all the experiments in this part to be well controlled and with clear results. For the community working with these genes it is quite useful to know that sug17 mutant allele has this second site mutation in Naca, and it could lead to the reinterpretation of the studies that used this mutant allele. It would be nice, but not essential, if the authors could make more progress to clarify the essential factors on the 3UTR of Naca that are disrupted by the roo insertion, and how this leads to lower levels of Naca transcript.

Then the authors follow up to unravel the molecular mechanisms downstream of Naca to cause the apoptosis inducing phenotype. They follow a candidate gene approach testing genes (JNK, P53) that are known to be involved in cell death/apoptosis induction. I think the exact molecular details remain elusive. This Naca hypomorphic mutation causes ER stress and impairs cytosolic proteostasis. Perhaps the authors could repeat these experiments with the bicaudal (Nacb) RNAi, to nail down that all these phenotypes are related to NAC and are not associated, perhaps, with a more elusive and mysterious

function of Naca (which could also be interesting). Overall, I liked the paper, although I remain puzzled with some of the findings, such as the P53 involvement, which is required but not sufficient to induce cell death in the fat body.

Reviewer #3 (Remarks to the Author):

NCOMMS-23-09819

The NAC complex (the nascent polypeptide-associated complex) comprises Nac² protein that binds basic transcription factor 3 (BTF3). NAC binds to nascent proteins that lack a signal peptide as they emerge from the ribosome, to block interaction with the signal recognition particle and ensure against mistranslocation of proteins to ER. As such NAC and NAC components play a critical role in proteostasis and ER stress. In this manuscript Takayuki Yamada et al. show that in *Drosophila* Nac² protects larval fat body (a liver and adipose-like tissue in insects) from cell death. They provide evidence suggesting that in *Drosophila* Nac² mutant affects ER integrity and proteasomal degradation in larval fat body resulting in disruption of proteostasis, JNK and p53 activation, that promote caspase activation. Although the study is not conceptually novel and the results are not unexpected, such role of Nac² protein has not been explored previously in the *Drosophila* larval fat body, that provides a nice model for in vivo studies. The fat body comprises polyploid cells and the authors show that this tissue is relatively resistant to apoptosis in response to cellular stress, including the ER stress. However, loss of Nac² results in sporadic apoptosis, apparently specifically in fat body cells.

Specific queries/comments:

1. Throughout the study the authors used cleaved Dcp1 staining as a surrogate marker for apoptosis, even though Dcp1 has limited function in apoptosis. They should use more robust methods for apoptosis (e.g. activation of Dronc, Drice and TUNEL). Cleaved caspase-3 antibody (used in some experiments that detects active Drice, is probably better than the cleaved Dcp1 immunostaining)
2. In addition to the overexpression of Rpr, Hid and Grim, authors should test if depletion of Diap1 results in caspase activation and apoptosis in the fat body, given that Diap1 is the main brake that keeps in check the activation of Dronc and Drice and the execution of apoptosis.
3. In Fig 3G, the authors suggest that apoptotic cells in fat body undergo secondary necrosis (based on PI staining). I assume they mean that apoptotic bodies generated by dead cells are degraded by secondary

necrosis (as is the case for cells in culture). Is PI binding to released extracellular DNA? Do they see similar staining when they induce apoptosis by Rpr, Hid or Grim overexpression (in Fig 1)?

4. In Fig 4, could the timing of apoptosis in fat body be dependent on the levels of proapoptotic proteins (e.g. Rpr, Hid, Grim, Dronc, Dark and Drice) or repression of Diap1? Several of these genes are known to be regulated by ecdysone during larval fat body development. Do Naca²² expression levels change during larval fat body development?

5. In Suppl Fig 6F, there does not appear to be much difference in ubiquitination levels between control and Naca-RNAi. Also, in the CBB protein staining image below the blot, it seems that Naca-RNAi results in increased protein degradation. Were these samples prepared in the presence of MG132? Given that in Fig 7F, only a few cells show proteasome activity, it is difficult to conclude that Naca is regulating the proteasome function.

6. Throughout the manuscript authors say that polyploidy generally makes cells resistant to apoptosis. For example, in Discussion they state "In mammals, polyploid cells are known to be resistant to DNA damage and apoptosis. Importantly, cancer cells often have complex karyotypes, and some cells have elevated genomic content and become polyploid tumor cells. Polyploidy facilitates tumor evolution and the acquisition of apoptosis-resistance characteristics in multiple incurable cancers." This is not entirely correct. Mammalian liver is the main tissue that comprises polyploid cells and polyploidy in liver is well known to be a tumor suppressive mechanism. Also, hepatocytes and other liver cells are not particularly resistant to apoptosis induced by cellular stress. While aneuploidy is a characteristic of tumor cells, polyploidy does not necessarily drive resistance to apoptosis or tumorigenesis.

Response to reviewers' comments.

Reviewer #1 (Remarks to the Author):

In this manuscript, the authors report that Nascent polypeptide-associated complex protein alpha subunit (NACa) is essential for protein homeostasis in *Drosophila* fat body cells. Specifically, the authors use independent genetic conditions (different alleles and RNAi lines) to show that even a mild loss of NACa activates cell death in the fat body. Stronger loss of function alleles interfere with cell survival in other tissues. They further show that Naca deficient fat body cells activate various stress response pathways, including p53, JNK, ER stress response, ROS response.

As the authors cite, Nac had been established as a critical regulator of cellular homeostasis. Along these lines, a recent *Drosophila* study (not cited by the authors) showed that Naca RNAi interferes with (heart) tissue development (Schroeder et al., PLoS Genetics PMID 36240221). The current manuscript does a fine job of describing the phenotypes of Naca loss. But given our existing knowledge of Nac, the overall message here appears somewhat of an expected outcome. The induction of apoptosis, in particular, involves the usual suspects such as p53, JNK, and RHG genes. Could the authors provide some previously unknown conceptual insights into the standard phenotypic description? How about adding new mechanistic insights regarding why fat body cells are more vulnerable to a mild loss of Naca? Alternatively, how about demonstrating whether ER stress (or possibly other sources of stress) is the leading “cause” of cell death induction in the fat body?

We cited the PLoS Genetics paper by Schroeder et al. in the Discussion section of the original manuscript. Importantly, this study showed that the effect of the *Naca* knockdown phenotype on heart tissue development is independent of cell death. Rather, it is caused by the misexpression of the Hox gene (Schroeder et al., 2022).

The reviewer has raised several suggestions to improve the depth of the study. Specifically, we addressed the causality between proteotoxic stress and cell death by using a chemical chaperone and reducing overall protein synthesis. In addition, we performed genetic interaction experiments

with cytoplasmic ribosomal protein genes to further demonstrate that proteotoxic stress (i.e., defective nascent peptides) is the leading cause of cell death induction. These results are discussed in detail below (**the response 4**).

In the revised version, we show the developmental changes of *NacA* and pro-apoptotic gene expression in the fat body (**see also the response 4 to reviewer #3**). Moreover, we show that ER chaperones are up-regulated, peaking at the mid-third instar during normal development (**Supplementary Fig. 3e**), suggesting an increased demand for protein folding in the fat body. These results further highlight that the initiation of apoptosis in *NacA* mutants is likely related to the unique protein synthesis activity in the fat body.

Technically, the authors have done an excellent job with proper controls and quantification of data. Below are points that could help enhance the conceptual depth of this manuscript for the authors' consideration:

1. It would be helpful to the readers if the authors introduced *Nac* in the Introduction. Specifically, a brief summary of what is already known about the *Nac* in cellular homeostasis and what are the critically unanswered questions would be helpful. As it is, the background information is sporadically introduced in the Results section, and the outstanding questions about the *Nac* complex are not well spelled out.

We described the NAC in the Introduction section accordingly.

2. The authors use the term "apoptosis" loosely throughout the manuscript. That term is based on morphological features that include cell-corpse engulfment. Since most fat body cells do not undergo engulfment, it is probably better to use the term "caspase-induced cell death."

Thank you for your comment. A hallmark of an apoptotic cell is the exposure of phosphatidylserine (PS) as an eat-me signal for engulfment (Nagata et al., 2010). To clarify PS exposure in fat body cells, we conducted Annexin V-FITC staining and found that some *NacA* mutant fat body cells were Annexin V/PI-double positive (**Fig. 3g**). In the revised version, we also demonstrated that many of the caspase-positive cells were positive for TUNEL (**Fig. 3h**), indicating that DNA fragmentation does occur (**see also the response 1 to reviewer #3**). Because fat body cells are polyploid and larger than macrophages, we speculate that secondary necrosis (i.e., the formation of apoptotic bodies) is a prerequisite for engulfment. Although it remains to

be determined whether apoptotic bodies are ultimately engulfed by macrophages, we believe that the term "apoptosis" is appropriate in this context.

3. In Figure 6, the authors detect UPR signaling markers (such as spliced Xbp1) after Naca loss. Based on this, the authors imply that apoptosis is induced by ER stress (line 291, 292). However, there is no experiment here establishing “causality” of ER stress. Moreover, the authors show in Figure 7I that cytoplasmic chaperones are also induced. It remains unclear whether the ER stress markers merely correlate with cell death, or indicate causality.

It has been well established that the NAC heterodimer directly binds to both the ribosome and the signal recognition particle (SRP) for ER targeting (Jomaa et al., 2022). Thus, the primary function of NAC is to maintain protein homeostasis in both the ER and the cytoplasm. Based on the known functions of NAC, it is reasonable to assume that disrupted proteostasis is the leading cause of caspase activation and cell death. Consistent with this, we demonstrated that Xbp1-GFP was up-regulated in the early third instar of the *Naca* knockdown larvae (**Fig. 6a, b**) when the cDcp1 signal was not yet globally elevated (**Fig. 4c, d**), suggesting that ER stress increased before caspase activation. We addressed causality through additional experiments, as described below (**the response 4**).

4. One way to enhance the depth of the study is to demonstrate what is the cause of cell death induction in Naca mutants. Can the authors suppress the fat body cell death phenotype by enhancing protein misfolding in the ER or the cytoplasm? Or alternatively, can they suppress the phenotype by reducing overall protein synthesis (thereby reducing defective nascent peptides)?

Thank you for raising this important issue and providing suggestions. To test whether enhanced protein folding suppresses fat body apoptosis, we used the chemical chaperone, 4-phenylbutyrate (4-PBA), which prevents the accumulation of unfolded proteins in the ER. Based on previous studies in *Drosophila* (Debattisti et al., 2014; Kang et al., 2002), 15 mM 4-PBA was used for transient feeding experiments to avoid potential chronic effects, as 4-PBA is also known as a histone deacetylase (HDAC) inhibitor. Feeding larvae with 4-PBA partially suppressed starvation-induced apoptotic progression (**Fig. 9a-c**), supporting the idea that ER stress is a cause of cell death. We also reduced translation initiation using rapamycin, a potent and specific inhibitor of mTor. Rapamycin strongly suppressed starvation-induced apoptotic progression, as evaluated by cDcp1 staining and nuclear condensation/fragmentation. Consistently, the expression of a dominant-negative form of mTor (*mTor-DN*) largely suppressed apoptotic induction in *Naca* knockdown larvae (**Fig. 9d, e**). Inhibition of mTor activity itself noticeably

reduced the nuclear and cell size in the fat body (**Supplementary Fig. 8a, b**), suggesting reduced protein translation.

To directly investigate whether reducing overall protein synthesis attenuates cell death phenotype, we focused on *Minute* mutants, a series of heterozygous mutants of a ribosomal protein gene (Brehme, 1939; Lambertsson, 1998). *Minute* mutants show a significant delay in their larval period, but adult flies are essentially normal. Recent structural analysis revealed that Nac β binds to RpL22 and RpL19, whereas Nac α does not bind directly to the ribosome (Jomaa et al., 2022). Because no *Minute* mutants have been reported for the *RpL22* gene (Marygold et al., 2007), we used two *Minute* mutants, *RpL19* and *RpS26*. Heterozygous *Minute* mutants (*RpL19/+* and *RpS26/+*) did not induce apoptosis in the fat body (**Fig. 9f**), whereas a half-dose reduction in ribosome function caused apoptosis in developing wing discs (**Supplementary Fig. 8c, d**), as reported previously (Akai et al., 2021; Baumgartner et al., 2021; Recasens-Alvarez et al., 2021). Under these conditions, *Minute* mutants partially suppressed fat body apoptosis caused by Nac α dysfunction (**Fig. 9g, h**). In contrast, the increased apoptosis in *Nac α* mutant wing discs was not altered or even exacerbated in the heterozygous *Minute* mutant background (**Supplementary Fig. 8c, d**). Collectively, these results strongly suggested that proteotoxic stress was the leading cause of cell death induction.

It should be noted that in wing discs, ribosomal protein deficiency induces proteotoxic stress and JNK activation, resulting in apoptosis (Akai et al., 2021; Baumgartner et al., 2021; Recasens-Alvarez et al., 2021). However, we showed that a stoichiometric imbalance in ribosomal proteins did not induce apoptosis in the fat body (**Fig. 9f**), which is consistent with our observation that proteasome disruption and JNK activation did not induce apoptosis in this tissue (**Fig. 1e**). Thus, we speculate that reducing NAC function is likely to cause more deleterious cellular stress in the fat body than that caused by ribosomal protein deficiency. Given that ribosome deficiency protects against ER stress in yeast (Steffen et al., 2012), the beneficial effects of ribosome deficiency outweigh any toxic effects, resulting in the prevention of apoptosis in the *Nac α* mutant background.

5. Figure 6G: p35 does not fully suppress the wing phenotype. The authors conclude that Nac α loss disrupts normal wing development irrespective of cell death. This conclusion could be misleading because p35 may not fully block apoptosis (I point out that the authors have not examined apoptosis markers in pupal stages). Also, p35 does not block initiator caspases such as Dronc (Meier et al. EMBO J. PMID 10675329), and therefore, it remains possible that the residual Dronc activity is causing the wing phenotype.

We agree with your comment. We have revised the text to reflect the potential that residual Dronc activity causes the wing phenotype.

6. The authors demonstrate “causality” of ROS in Figure 8A by overexpressing ROS quenchers such as *Cat* or *Sod2*. This is excellent. Could they use this assay to demonstrate that ROS is a cause of p53 and JNK induction?

To address your suggestion, we analyzed p53 and JNK signaling in the fat body expressing *Cat* by RT-qPCR. Consistent with the cell death phenotype, *Cat* expression partially suppressed *rpr* and *puc* expression in *Nac α* mutants (**Fig. 8c**). Notably, *Cat* expression had little effect on the expression of ER chaperone gene, suggesting that ROS mainly acts downstream of ER stress to boost p53 and JNK activation.

7. What happens if one overexpresses *Nac α* in the fat body. Would that protect those cells from adverse conditions?

We performed an overexpression analysis of *Nac α* , but did not observe a visible phenotype in any tissue during development. One possible reason why *Nac α* overexpression had no phenotype may be due to the high level of endogenous *Nac α* expression ("very high expression" based on the FlyAtlas expression data), which is comparable to ribosomal gene expression.

Reviewer #2 (Remarks to the Author):

Yamada et al start by observing that the *sug17* mutant allele has a second site mutation that causes small nuclei/apoptosis in the fat body. They pursue this to discover a hypomorphic mutation in the *Naca* gene, encoding part of the heterodimer that binds to the nascent ribosome-associated polypeptides, caused by a *roo* insertion in the 3'UTR of *Naca*, causing a reduction in the transcription of *Naca*. I find all the experiments in this part to be well controlled and with clear results. For the community working with these genes it is quite useful to know that *sug17* mutant allele has this second site mutation in *Naca*, and it could lead to the reinterpretation of the studies that used this mutant allele. It would be nice, but not essential, if the authors could make more progress to clarify the essential factors on the 3'UTR of *Naca* that are disrupted by the *roo* insertion, and how this leads to lower levels of *Naca* transcript.

Thank you for your comments. To understand how the *roo* insertion in the 3'UTR leads to lower levels of *Nac α* transcript, we performed RT-qPCR analysis using primer sets targeting different

Naca mRNA regions (**Supplementary Fig. 2e**). As shown in the original version of the manuscript, *Naca* transcript levels decreased to 40% when the common ORF region and spliced transcripts were analyzed. *Naca* transcript levels decreased to 1% when a 3'UTR region was analyzed, suggesting that the *roo* insertion prevents *Naca* transcription beyond the insertion. Interestingly, we found increased *Naca* transcript levels when analyzing a 5'UTR region. Thus, the *roo* element insertion likely prevents transcriptional progression and/or maturation rather than transcriptional initiation in *Naca^l* mutants.

Then the authors follow up to unravel the molecular mechanisms downstream of *Naca* to cause the apoptosis inducing phenotype. They follow a candidate gene approach testing genes (JNK, P53) that are known to be involved in cell death/apoptosis induction. I think the exact molecular details remain elusive. This *Naca* hypomorphic mutation causes ER stress and impairs cytosolic proteostasis. Perhaps the authors could repeat these experiments with the bicaudal (*Nacb*) RNAi, to nail down that all these phenotypes are related to NAC and are not associated, perhaps, with a more elusive and mysterious function of *Naca* (which could also be interesting).

Thank you for pointing this out. To address this, we analyzed the *bic/Nacj3* mutant phenotype in detail. In the original version of the manuscript, we showed that *bic* knockdown resulted in apoptosis in the fat body (**Supplementary Fig. 1f**). In the revised manuscript, we have shown that many *Naca* mutant phenotypes are recapitulated in *bic* mutants, as summarized below:

- 1) In addition to *bic* knockdown, the *P*-element insertion allele of *bic* showed caspase activation in the fat body (**Supplementary Fig. 1g, h**).
- 2) *bic* knockdown increased ER stress in the fat body, as assessed by Xbp1-GFP and Xbp1-target gene expression (**Supplementary Fig. 5a, b**).
- 3) *bic* knockdown increased JNK signaling and the expression of p53 and pro-apoptotic genes in the fat body (**Supplementary Fig. 5c**).
- 4) *bic* knockdown increased ubiquitin signals in the fat body, suggesting impaired cytosolic proteostasis (**Supplementary Fig. 6h, i**).

Importantly, we have shown in the revised manuscript that two heterozygous *Minute* mutants (*RpL19/+* and *RpS26/+*) alleviated fat body apoptosis in *Naca* mutants (**see also the response 4 to reviewer #1**). RpL19 is a ribosomal subunit that binds to *bic/Nacj3* (Jomaa et al., 2022). Taken together, we believe that the fat body apoptosis observed in the *Naca* mutants was primarily caused by the reduction of NAC composed of *Naca* and *bic/Nacj3*.

Overall, I liked the paper, although I remain puzzled with some of the findings, such as the P53 involvement, which is required but not sufficient to induce cell death in the fat body.

As discussed in the original manuscript, we think that chromatin silencing of the p53 target locus is one reason why p53 is not sufficient to induce cell death, as described previously (Zhang et al., 2014). How *Naca* mutants overcome chromatin silencing of pro-apoptotic genes remains to be elucidated. Other signaling pathways that drive apoptosis in the fat body will be our next research objective.

Reviewer #3 (Remarks to the Author):

The NAC complex (the nascent polypeptide-associated complex) comprises *Naca* protein that binds basic transcription factor 3 (BTF3). NAC binds to nascent proteins that lack a signal peptide as they emerge from the ribosome, to block interaction with the signal recognition particle and ensure against mistranslocation of proteins to ER. As such NAC and NAC components play a critical role in proteostasis and ER stress. In this manuscript Takayuki Yamada et al. show that in *Drosophila* *Naca* protects larval fat body (a liver and adipose-like tissue in insects) from cell death. They provide evidence suggesting that in *Drosophila* *Naca* mutant affects ER integrity and proteasomal degradation in larval fat body resulting in disruption of proteostasis, JNK and p53 activation, that promote caspase activation. Although the study is not conceptually novel and the results are not unexpected, such role of *Naca* protein has not been explored previously in the *Drosophila* larval fat body, that provides a nice model for in vivo studies. The fat body comprises polyploid cells and the authors show that this tissue is relatively resistant to apoptosis in response to cellular stress, including the ER stress. However, loss of *Naca* results in sporadic apoptosis, apparently specifically in fat body cells.

Specific queries/comments:

1. Throughout the study the authors used cleaved Dcp1 staining as a surrogate maker for apoptosis, even though Dcp1 has limited function in apoptosis. They should use more robust methods for apoptosis (e.g. activation of Dronc, Drice and TUNEL). Cleaved caspase-3 antibody (used in some experiments that detects active Drice, is probably better than the cleaved Dcp1 immunostaining.

Thank you for your insightful comments. In our study, we first used a cleaved caspase-3 antibody from CST (#9661); however, the new batch did not work properly. Therefore, we used the cDcp1 antibody from CST (#9578). We would like to mention that these antibodies do not specifically detect active Drice or Dcp1. The cleaved caspase-3 antibody recognizes cleaved Drice, Dcp1, and unknown substrate(s) that may be involved in the non-apoptotic functions of Dronc (Fan and

Bergmann, 2010). The cDcp1 antibody recognizes both cleaved Dcp1 and Drice (Li et al., 2019). We detected the expression of both *Dcp1* and *Drice* in the fat body, as described below (**the response 4**). Whether fat body apoptosis in *Nac2* mutants is driven by Dcp1 and/or Drice is beyond the scope of the present study.

To further characterize the apoptotic phenotype in the fat body, we have shown in the revised manuscript that cDcp1-positive fat body cells in *Nac2* mutants are positive for the fluorescent reporter GC3Ai, which responds to both Dcp1 and Drice (Schott et al., 2017) (**Fig. 3i**). We also showed that most cDcp1-positive fat body cells in *Nac2* mutants were TUNEL-positive (**Fig. 3h**). Therefore, we believe that cDcp1 staining can be used as a surrogate marker of apoptosis in the fat bodies.

2. In addition to the overexpression of Rpr, Hid and Grim, authors should test if depletion of Diap1 results in caspase activation and apoptosis in the fat body, given that Diap1 is the main brake that keeps in check the activation of Dronc and Drice and the execution of apoptosis.

Accordingly, we analyzed the involvement of Diap1. The transient knockdown of *Diap1* resulted in caspase activation and apoptosis in the fat body (**Supplementary Fig. 1a**). This phenotype was similar to that observed with Rpr, Hid, and Grim overexpression. Changes in *Diap1* expression during development are discussed below (**the response 4**).

3. In Fig 3G, the authors suggest that apoptotic cells in fat body undergo secondary necrosis (based on PI staining). I assume they mean that apoptotic bodies generated by dead cells are degraded by secondary necrosis (as is the case for cells in culture). Is PI binding to released extracellular DNA? Do they see similar staining when they induce apoptosis by Rpr, Hid or Grim overexpression (in Fig 1)?

To address the reviewer's questions, we performed PI staining of Rpr-, Hid-, or Grim-overexpressing fat bodies. The overexpression of *rpr*, *hid*, or *grim* led to PI-positive secondary necrosis (**Supplementary Fig. 1b**). Consistently, knockdown of *Diap1* resulted in similar PI-positive signals. These results suggest that secondary necrosis is not specific to *Nac2* mutants, but is a general feature of apoptotic fat body cells. We also showed that some *Nac2* mutant fat body cells were Annexin V/PI double-positive (**Fig. 3g**) (**see the response to comment 2 from reviewer #1**).

It is difficult to distinguish between intracellular and extracellular DNA with PI staining. Theoretically, PI binds to the released extracellular DNA. However, we believe that the observed

PI signals most likely originated from intracellular DNA and RNA, although the plasma membranes were damaged.

4. In Fig 4, could the timing of apoptosis in fat body be dependent on the levels of proapoptotic proteins (e.g. Rpr, Hid, Grim, Dronc, Dark and Drice) or repression of Diap1? Several of these genes are known to be regulated by ecdysone during larval fat body development. Do *Nac2* expression levels change during larval fat body development?

We agree with the reviewer that this is an important issue. Accordingly, we analyzed *Nac2* expression levels during larval fat body development. *Nac2* was almost constantly expressed during the larval stages, whereas *bic/Nac2* was gradually decreased (**Fig. 4g and Supplementary Fig. 3e**). Although caspase activation was detectable during the second instar, we did not observe any changes in the expression of anti-apoptotic *Diap1* or pro-apoptotic genes (*rpr*, *hid*, *grim*, *Dcp1*, and *Drice*) between the early second and the early third instars. Thus, the timing of apoptosis in the fat body is independent of changes in the expression of anti-apoptotic and pro-apoptotic genes. We failed to detect reliable signals for *Dronk* and *Dark* in the larval fat body by qRT-PCR with three different primer sets.

As the reviewer mentioned, all of these genes were up-regulated in the late third instar, most likely by ecdysone signaling. Since the up-regulation of *EcR* and *br*, which are targets of ecdysone signaling, was already detectable in the early third instar, anti-apoptotic and pro-apoptotic genes appeared to respond to high ecdysteroid titers in the late third instar. We extensively analyzed the involvement of ecdysteroids in apoptosis induction because we initially hypothesized that ecdysone signaling might trigger apoptosis. We have previously shown that *EcR* inhibition has no effect on fat body apoptosis in *Nac2* mutants (**Fig. 8a, b**). In the revised manuscript, we showed that feeding larvae with 20-hydroxyecdysone failed to accelerate apoptotic progression in *Nac2* knockdown fat bodies (**Supplementary Fig. 7a-c**). Collectively, these results suggested that ecdysone signaling was not critically involved in fat body apoptosis.

Expression analysis by RT-qPCR provides quantitative information at the tissue level. However, the cellular heterogeneity remains to be determined. Given that loss of *Diap1* results in fat body apoptosis, we analyzed the expression pattern of *Diap1* using a *Diap1-lacZ* reporter strain. LacZ expression was almost uniformly detected in fat body cells (**Supplementary Fig. 1c**), suggesting that the fluctuation in *Diap1* expression was not responsible for the apoptotic cell fate in a mosaic manner. We also analyzed whether repression of *Diap1* affected the apoptotic phenotype of *Nac2* mutants. Since *Diap1* mutants are lethal in embryos and/or early larvae, we analyzed *Nac2* mutant larvae in a *Diap1* heterozygous background. A half-dose reduction in

Diap1 promoted apoptotic progression in the early third instar, whereas the timing of fat body apoptosis onset was not altered (**Fig. 4h and Supplementary Fig. 3f, g**).

Interestingly, the expression of the ER chaperones, *Hsc70-3* and *Pdi*, increased during development, peaking at the mid-third instar (**Supplementary Fig. 3e**), suggesting an increased demand for protein folding in the fat body. In the revised manuscript, we have shown that reduced protein synthesis ameliorates fat body apoptosis in *Nac2* mutants (**see also the response 4 to reviewer #1**). These results suggest that the initiation of apoptosis in *Nac2* mutants is mainly related to increased protein synthesis, although *Diap1* indeed acts as a brake on the execution of apoptosis in the fat body.

5. In Suppl Fig 6F, there does not appear to be much difference in ubiquitination levels between control and *Naca*-RNAi. Also, in the CBB protein staining image below the blot, it seems that *Naca*-RNAi results in increased protein degradation. Were these samples prepared in the presence of MG132? Given that in Fig 7F, only a few cells show proteasome activity, it is difficult to conclude that *Naca* is regulating the proteasome function.

We apologize for the lack of an explanation of the CBB image (**Supplementary Fig. 6f**). Well-defined bands at 80kDa in the control samples are the larval serum proteins (*Lsp1/2*), which considerably increase in the fat body during the third instar in response to ecdysteroids (Powell et al., 1984). The absence of these bands in *Nac2* mutants suggests that *Nac2* mutants fail to respond correctly to ecdysteroids or synthesize these secreted proteins due to functional impairment of the ER. We discussed these issues in the Supplementary figure's legend. The samples were prepared in the absence of MG132.

Regarding the fluorescence probe for proteasome activity (**Fig. 7e in the revised version**), we have provided its quantification in **Fig. 7f**. Although there are huge variations in proteasome probe signals between cells, *Nac2* mutants showed noticeably reduced fluorescence intensity. To further strengthen the manuscript, we now show that *Nac2* mutant fat body cells increase ubiquitin signals, as revealed by immunostaining (**Fig. 7h and Supplementary Fig. 6j**). Consistently, the knockdown of *Nac2* and *bic* strongly increased ubiquitin signals and frequently showed ubiquitin-positive aggregates in the cytoplasm (**Supplementary Fig. 6h, i**). These results further supported the idea that a reduction in NAC function impairs proteasome function.

Notably, misfolded proteins in the ER are retro-translocated to the cytoplasm for ubiquitin-mediated degradation, which is referred to as the ER-associated degradation (ERAD) pathway (Brodsky, 2012). In addition, a recent study revealed that NAC recruits an enzyme for N-terminal methionine excision, a conserved process that ensures the stability and folding of the cytosolic

proteins, to the ribosome (Gamerding et al., 2023). Therefore, it is expected that the reduction in NAC function will impact not only the ER, but also cytoplasmic proteasome activity.

6. Throughout the manuscript authors say that polyploidy generally makes cells resistant to apoptosis. For example, in Discussion they state "In mammals, polyploid cells are known to be resistant to DNA damage and apoptosis. Importantly, cancer cells often have complex karyotypes, and some cells have elevated genomic content and become polyploid tumor cells. Polyploidy facilitates tumor evolution and the acquisition of apoptosis-resistance characteristics in multiple incurable cancers." This is not entirely correct. Mammalian liver is the main tissue that comprises polyploid cells and polyploidy in liver is well known to be a tumor suppressive mechanism. Also, hepatocytes and other liver cells are not particularly resistant to apoptosis induced by cellular stress. While aneuploidy is a characteristic of tumor cells, polyploidy does not necessarily drive resistance to apoptosis or tumorigenesis.

Thank you for your comments on improving the manuscript. We agree that polyploidy generally renders cells resistant to apoptosis. We carefully revised the manuscript and specified the cell type and stresses to accurately describe the relationship between polyploidy and apoptosis.

References

- Akai, N., Ohsawa, S., Sando, Y. and Igaki, T. (2021). Epithelial cell-turnover ensures robust coordination of tissue growth in *Drosophila* ribosomal protein mutants. *PLoS Genet.* 17, e1009300.
- Baumgartner, M. E., Dinan, M. P., Langton, P. F., Kucinski, I. and Piddini, E. (2021). Proteotoxic stress is a driver of the loser status and cell competition. *Nat. Cell Biol.* 23, 136-146.
- Brehme, K. S. (1939). A Study of the Effect on Development of "Minute" Mutations in *Drosophila Melanogaster*. *Genetics* 24, 131-161.
- Brodsky, J. L. (2012). Cleaning up: ER-associated degradation to the rescue. *Cell* 151, 1163-1167.
- Debattisti, V., Pendin, D., Ziviani, E., Daga, A. and Scorrano, L. (2014). Reduction of endoplasmic reticulum stress attenuates the defects caused by *Drosophila* mitofusin depletion. *J. Cell Biol.* 204, 303-312.
- Fan, Y. and Bergmann, A. (2010). The cleaved-Caspase-3 antibody is a marker of Caspase-9-like DRONC activity in *Drosophila*. *Cell Death Differ* 17, 534-539.
- Gamerding, M., Jia, M., Schloemer, R., Rabl, L., Jaskolowski, M., Khakzar, K. M., Ulusoy, Z., Wallisch, A., Jomaa, A., Hunaeus, G., et al. (2023). NAC controls cotranslational N-terminal methionine excision in eukaryotes. *Science* 380, 1238-1243.
- Jomaa, A., Gamerding, M., Hsieh, H. H., Wallisch, A., Chandrasekaran, V., Ulusoy, Z., Scaiola, A., Hegde,

- R. S., Shan, S. O., Ban, N., et al. (2022). Mechanism of signal sequence handover from NAC to SRP on ribosomes during ER-protein targeting. *Science* 375, 839-844.
- Kang, H. L., Benzer, S. and Min, K. T. (2002). Life extension in *Drosophila* by feeding a drug. *Proc. Natl. Acad. Sci. U. S. A.* 99, 838-843.
- Lambertsson, A. (1998). The minute genes in *Drosophila* and their molecular functions. *Adv. Genet.* 38, 69-134.
- Li, M., Sun, S., Priest, J., Bi, X. and Fan, Y. (2019). Characterization of TNF-induced cell death in *Drosophila* reveals caspase- and JNK-dependent necrosis and its role in tumor suppression. *Cell Death Dis.* 10, 613.
- Marygold, S. J., Roote, J., Reuter, G., Lambertsson, A., Ashburner, M., Millburn, G. H., Harrison, P. M., Yu, Z., Kenmochi, N., Kaufman, T. C., et al. (2007). The ribosomal protein genes and Minute loci of *Drosophila melanogaster*. *Genome Biol.* 8, R216.
- Nagata, S., Hanayama, R. and Kawane, K. (2010). Autoimmunity and the clearance of dead cells. *Cell* 140, 619-630.
- Powell, D., Sato, J. D., Brock, H. W. and Roberts, D. B. (1984). Regulation of synthesis of the larval serum proteins of *Drosophila melanogaster*. *Dev. Biol.* 102, 206-215.
- Recasens-Alvarez, C., Alexandre, C., Kirkpatrick, J., Nojima, H., Huels, D. J., Snijders, A. P. and Vincent, J. P. (2021). Ribosomopathy-associated mutations cause proteotoxic stress that is alleviated by TOR inhibition. *Nat. Cell Biol.* 23, 127-135.
- Schott, S., Ambrosini, A., Barbaste, A., Benassayag, C., Gracia, M., Proag, A., Rayer, M., Monier, B. and Suzanne, M. (2017). A fluorescent toolkit for spatiotemporal tracking of apoptotic cells in living *Drosophila* tissues. *Development* 144, 3840-3846.
- Schroeder, A. M., Nielsen, T., Lynott, M., Vogler, G., Colas, A. R. and Bodmer, R. (2022). Nascent polypeptide-Associated Complex and Signal Recognition Particle have cardiac-specific roles in heart development and remodeling. *PLoS Genet.* 18, e1010448.
- Steffen, K. K., McCormick, M. A., Pham, K. M., MacKay, V. L., Delaney, J. R., Murakami, C. J., Kaerberlein, M. and Kennedy, B. K. (2012). Ribosome deficiency protects against ER stress in *Saccharomyces cerevisiae*. *Genetics* 191, 107-118.
- Zhang, B., Mehrotra, S., Ng, W. L. and Calvi, B. R. (2014). Low levels of p53 protein and chromatin silencing of p53 target genes repress apoptosis in *Drosophila* endocycling cells. *PLoS Genet.* 10, e1004581.

REVIEWERS' COMMENTS

Reviewer #1 (Remarks to the Author):

The original manuscript was technically excellent, and the main novelty of the manuscript was in the validation of the expected role of Nac in the *Drosophila* fat body. Somewhat interesting was the observation that the fat body was particularly vulnerable to the loss of Nac. The authors have addressed many of my original suggestions in this revised manuscript. Still, there are a couple of points that the authors have not fully addressed:

1. In this revised manuscript, the authors further support their point that proteotoxic stress is the cause of cell death in Nac mutant tissues. I agree with that point. But is ER stress the primary source of that proteotoxic stress? The authors do use 4-PBA treatment experiments to make their case. It is true that the 4-PBA reportedly acts as a chemical chaperone. But as the authors acknowledge, it also acts as an inhibitor of HDAC. Therefore, the 4-PBA treatment experiment is insufficient to establish the causal role of ER stress in the author's model. I suggest either deleting conclusions related to ER stress playing a causal role or showing additional experimental evidence for the role of ER stress.

2. The authors conclude that the cell death they see in the fat body is a form of apoptosis. I had pointed out that apoptosis is defined as a morphological form of cell death that involves cell corpse engulfment. The authors responded by writing that "although it remains to be determined whether apoptotic bodies are ultimately engulfed by macrophages, we believe that the term apoptosis is appropriate in this context." I am afraid I have to disagree here. The authors may want to read about the original definition of apoptosis, which was based on morphological features that include cell corpse engulfment.

<https://pubmed.ncbi.nlm.nih.gov/4561027/>

It is always good practice to use the correct terms. Using more conservative terms, such as caspase-associated cell death (or even, just cell death), will not take anything away from the authors.

Reviewer #2 (Remarks to the Author):

I am happy with the revisions made by the authors in reply to my original comments and therefore, I recommend in favour of the publication of the manuscript.

Reviewer #3 (Remarks to the Author):

In the revised version the authors seem to have addressed my main concerns.

Response to reviewers' comments.

Our point-by-point responses are written below (in blue).

Reviewer #1 (Remarks to the Author):

The original manuscript was technically excellent, and the main novelty of the manuscript was in the validation of the expected role of Nac in the Drosophila fat body. Somewhat interesting was the observation that the fat body was particularly vulnerable to the loss of Nac. The authors have addressed many of my original suggestions in this revised manuscript. Still, there are a couple of points that the authors have not fully addressed:

1. In this revised manuscript, the authors further support their point that proteotoxic stress is the cause of cell death in Nac mutant tissues. I agree with that point. But is ER stress the primary source of that proteotoxic stress? The authors do use 4-PBA treatment experiments to make their case. It is true that the 4-PBA reportedly acts as a chemical chaperone. But as the authors acknowledge, it also acts as an inhibitor of HDAC. Therefore, the 4-PBA treatment experiment is insufficient to establish the causal role of ER stress in the author's model. I suggest either deleting conclusions related to ER stress playing a causal role or showing additional experimental evidence for the role of ER stress.

We have toned down the conclusions regarding the role of ER stress, emphasizing instead that proteotoxic stress plays a causal role in fat body cell death.

2. The authors conclude that the cell death they see in the fat body is a form of apoptosis. I had pointed out that apoptosis is defined as a morphological form of cell death that involves cell corpse engulfment. The authors responded by writing that "although it remains to be determined whether apoptotic bodies are ultimately engulfed by macrophages, we believe that the term apoptosis is appropriate in this context." I am afraid I have to disagree here. The authors may want to read about the original definition of apoptosis, which was based on morphological features that include cell corpse engulfment.

<https://pubmed.ncbi.nlm.nih.gov/4561027/>

It is always good practice to use the correct terms. Using more conservative terms, such as caspase-associated cell death (or even, just cell death), will not take anything away from the authors.

Accordingly, we changed the word throughout the manuscript and added a caveat stating that while the engulfment of cell corpses has not been shown the other experiments are consistent with apoptosis.